# Curriculum Reinforcement Learning for Black-Box Prompt Tuning via Large Language Models

**Shuai Gong** [1 2]   **Chaoran Cui** [1 2]   **Xiaolin Dong** [1 2]   **Chunyun Zhang** [1 2]   **Linwei Fan** [1 2]

## Abstract

Black-box prompt tuning (BBPT) aims to optimize input prompts for large models where internal parameters and gradients are inaccessible. However, existing methods fail to simultaneously address the dual challenges of prompt interpretability and query efficiency. To address these challenges, we propose CRL-BPT, a curriculum reinforcement learning framework that utilizes a large language model as an agent to generate human-readable prompts. Specifically, CRL-BPT implements a dynamic curriculum schedule on two auxiliary objectives: an *imitation loss* and an *innovation loss*. By dynamically weighting these objectives, CRL-BPT regularizes the RL process, guiding the agent from mimicking reference prompts to discovering novel patterns. Additionally, we introduce tailored stabilization mechanisms comprising *historical loss normalization* and *relative reward calibration* to promote more stable training. Extensive experiments demonstrate that CRL-BPT establishes new state-of-the-art performance and generates highly interpretable prompts under a strict budget of API calls. Code is available at `https://github.com/GongShuai8210/CRL-BPT`.

## 1. Introduction

Vision-language models (VLMs), such as CLIP (Radford et al., 2021), have attracted widespread attention due to their strong zero-shot capabilities. However, adapting large VLMs to downstream tasks poses considerable challenges, as fine-tuning all model parameters is computationally pro-

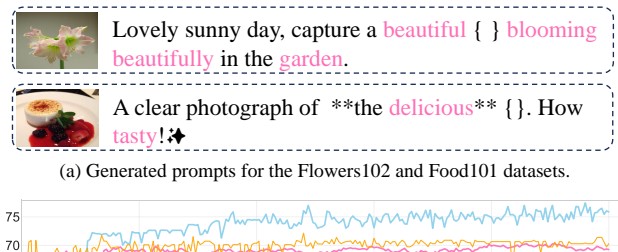

(a) Generated prompts for the Flowers102 and Food101 datasets.

(b) Accuracy against number of API queries on the Flowers102 dataset.

*Figure 1.* Illustration of CRL-BPT's interpretability and efficiency. (a) CRL-BPT synthesizes human-readable prompts with rich domain-specific semantics (e.g., "blooming beautifully" for flowers, and "tasty" for food). (b) It achieves superior accuracy and API efficiency, significantly outperforming baseline methods.

hibitive and impractical for scalable deployment. Drawing inspiration from natural language processing (NLP) techniques, prompt tuning (Zhou et al., 2022; Khattak et al., 2023; Cui et al., 2025a;b) has been proposed as an efficient method for adapting VLMs. Unlike full-model fine-tuning, prompt tuning optimizes a sequence of learnable prompt tokens as additional input while keeping the model parameters frozen.

Currently, powerful VLMs are generally released as Model-as-a-Service (MaaS) (Comanici et al., 2025) due to commercial and safety considerations. In this setting, users can only interact with models via API queries and cannot access internal parameters or gradients. Consequently, conventional gradient-based prompt tuning methods, such as CoOp (Zhou et al., 2022), are inapplicable to such black-box models.

To overcome this challenge, recent works (Sun et al., 2022; Meng et al., 2025) have explored black-box prompt tuning (BBPT). In the absence of gradients, existing BBPT methods (Yu et al., 2023; Park et al., 2025) predominantly focus on optimizing continuous embedding vectors, known as *soft prompts*, via derivative-free algorithms. However, these approaches suffer from two major limitations. First, derivative-free algorithms generally converge slowly in the high-dimensional parameter space of soft prompts, neces-

[1]School of Computing and Artificial Intelligence, Shandong University of Finance and Economics, Jinan, China, 250014 [2]Shandong Key Laboratory of Lightweight Intelligent Computing and Visualization for Digital Economy, Jinan, China, 250014. Correspondence to: Chaoran Cui <crcui@sdufe.edu.cn>.

*Proceedings of the $43^{rd}$ International Conference on Machine Learning*, Seoul, South Korea. PMLR 306, 2026. Copyright 2026 by the author(s).

sitating extensive API calls and incurring high query costs. Furthermore, the learned soft prompts inherently lack interpretability; they act as opaque numerical vectors that cannot be understood by human users.

To prioritize interpretability, several NLP works (Choi et al., 2024; Kwon et al., 2024) have proposed using a large language model (LLM) as an agent to generate human-readable discrete tokens, i.e., *hard prompts*. Given that gradients are inaccessible, these methods generally formulate prompt generation as a reinforcement learning (RL) problem (Sutton et al., 1998). However, searching the vast discrete token space without gradient guidance inevitably leads to blind exploration. RL agents typically suffer from a *cold start* phase, where they lack useful priors and therefore waste many queries on low-quality prompts (Deng et al., 2022). As a result, learning hard prompts remains burdened by prohibitive query costs.

To simultaneously achieve prompt interpretability and API efficiency, we propose Curriculum Reinforcement Learning for Black-box Prompt Tuning via LLMs (CRL-BPT). As illustrated in Fig. 1, CRL-BPT generates interpretable, semantically relevant prompts while achieving state-of-the-art (SOTA) performance with significantly fewer API queries than existing baselines. Specifically, CRL-BPT employs an LLM as a policy agent to generate human-readable discrete prompts, which is optimized via RL algorithm. To address the critical bottleneck of API efficiency, we incorporate a curriculum learning strategy that guides the agent's exploration gradually *from simple imitation to complex innovation*. We enforce this transition via a dynamic weighting scheme on two auxiliary loss terms: an *imitation loss* and an *innovation loss*. In the early stages, the imitation loss is assigned a high weight, encouraging the agent to mimic linguistic patterns from high-quality reference prompts, thereby minimizing blind exploration. As training progresses, this weight decays to prevent overfitting, while the influence of the innovation loss explicitly increases. This curriculum shift compels the agent to discover novel prompt structures distinct from previously generated prompts. By dynamically balancing these objectives, CRL-BPT effectively navigates the trade-off between exploiting prior knowledge and exploring new solutions, significantly reducing API costs.

To ensure stable training, CRL-BPT further incorporates tailored stabilization mechanisms for both the objective functions and reward signals. First, to balance the optimization landscape, we introduce *historical loss normalization*. Since the RL, imitation, and innovation terms exhibit distinct magnitudes, we normalize each term based on its historical moving average. This ensures that the gradient direction is effectively governed by the designed curriculum schedule rather than being biased by the scale of raw loss values. Second, to mitigate the high variance arising

from stochastic mini-batches, we propose *relative reward calibration*. Instead of relying on the raw reward derived from the agent-generated prompts, we calibrate the reward against a baseline prompt, scaled by a theoretically optimal coefficient. This design minimizes reward variance caused by fluctuating batch difficulties while maintaining unbiased policy updates.

In a nutshell, our main contributions are as follows:

- We propose CRL-BPT, a curriculum reinforcement learning framework for black-box prompt tuning via LLMs. By implementing a dynamic curriculum schedule that guides the agent from imitation to innovation, CRL-BPT generates interpretable prompts while significantly improving API efficiency.

- We introduce tailored stabilization mechanisms to promote more stable training. Specifically, we design historical loss normalization to align the scales of different objectives, and relative reward calibration to minimize the variance caused by stochastic mini-batches.

- Extensive experiments on benchmark datasets demonstrate that CRL-BPT achieves new SOTA performance. Our results validate the method's superiority in accuracy, efficiency, and interpretability, proving its effectiveness in budget-constrained black-box scenarios.

## 2. Related Work

### 2.1. Black-Box Prompt Tuning

Black-box prompt tuning (BBPT) aims to optimize input prompts for large models where internal parameters and gradients are inaccessible. Existing BBPT methods are generally categorized into continuous soft prompt tuning and discrete hard prompt tuning.

**Soft Prompt Tuning.** Early research in BBPT focused on optimizing continuous embedding vectors via derivative-free techniques. Pioneering works like BBT (Sun et al., 2022) and BlackVIP (Oh et al., 2023) demonstrated the feasibility of optimizing soft prompts without gradient access. For VLMs, BPTVLM (Yu et al., 2023) applied evolutionary strategies to evolve prompts, and ZIP (Park et al., 2025) employed zeroth-order optimization to estimate gradients. Despite their effectiveness, these algorithms generally suffer from slow convergence in high-dimensional parameter spaces, necessitating extensive API queries. Furthermore, the learned soft prompts inherently act as opaque vectors, lacking interpretability for human users.

**Hard Prompt Tuning.** To address the lack of interpretability, recent research has shifted towards optimizing discrete, human-readable prompt tokens. Several ap-

proaches (Choi et al., 2024; Kwon et al., 2024) utilize an LLM as an agent to generate these tokens. Given the gradient-free constraint, these methods typically formulate prompt generation as an RL problem (Sutton et al., 1998). However, applying RL to the vast discrete token space inevitably leads to blind exploration during the early stages, often referred to as the *cold start* problem (Deng et al., 2022). Agents waste thousands of interactions on low-quality prompts before discovering effective patterns, resulting in prohibitive API costs. Our CRL-BPT method addresses these challenges by incorporating a curriculum learning strategy that efficiently guides exploration, significantly reducing query overhead.

## 2.2. Curriculum Reinforcement Learning

Curriculum learning (Bengio et al., 2009) is a training strategy inspired by human cognitive development, where learning is structured in a meaningful order of increasing difficulty. By smoothing the optimization landscape, curriculum learning has been proven to accelerate convergence and improve generalization (Wang et al., 2021). In the realm of sequential decision-making, this paradigm is adapted as curriculum reinforcement learning (CRL). According to the survey by Narvekar et al. (2020), CRL methods broadly involve sequencing tasks or prioritizing specific experiences to guide agent learning.

Recently, CRL principles have been integrated into the optimization of LLMs, predominantly focusing on *sample-level approaches* (Lee et al., 2024; Team et al., 2025). These methods typically estimate data difficulty offline and sort training samples into a fixed easy-to-hard sequence. In contrast, our CRL-BPT method implements a *objective-based curriculum*. Instead of filtering or reordering data, we progressively shift the optimization objective—transitioning from mimicking reference prompts to exploring novel patterns. This approach allows the agent to effectively leverage prior knowledge in the early stages while continuously adapting to more complex exploration challenges.

## 3. Preliminaries

**Problem Formulation.** We consider the BBPT setting (Yu et al., 2023; Park et al., 2025) where the internal parameters and gradients of the VLM are inaccessible. Let $\mathcal{M}$ denote the black-box VLM (e.g., CLIP (Radford et al., 2021)). Given an input image $\boldsymbol{x}$ and a text prompt $\boldsymbol{z}$, the model outputs a probability distribution over class labels $y \in \mathcal{Y}$, denoted as $p(y|\boldsymbol{x}, \boldsymbol{z}) = \mathcal{M}(\boldsymbol{x}, \boldsymbol{z})$. The goal is to find an optimal discrete prompt $\boldsymbol{z}^*$ that maximizes the classification accuracy on a dataset $\mathcal{D}$.

**Hard Prompt Tuning with RL.** Following recent approaches (Kwon et al., 2024), we formulate discrete prompt

generation as a RL problem. We employ an LLM as the policy agent $\pi_\theta$, parameterized by $\theta$. The process is modeled as a Markov decision process:

- **State**: The agent receives a task context $\boldsymbol{s}$, which consists of meta-instructions describing the task goal (see Appendix C.2 for details).
- **Action**: Conditioned on $\boldsymbol{s}$, the agent generates a discrete prompt $\boldsymbol{z} \sim \pi_\theta(\cdot|\boldsymbol{s})$, which serves as the action applied to the black-box environment.
- **Reward**: The VLM evaluates $\boldsymbol{z}$ on a batch of images from $\mathcal{D}$, and the prediction performance constitutes the reward signal $R$.

Accordingly, the optimization objective is to maximize the expected reward:

$$J(\theta) = \mathbb{E}_{\boldsymbol{z} \sim \pi_\theta(\cdot|\boldsymbol{s})} \left[ \mathbb{E}_{(\boldsymbol{x},y) \sim \mathcal{D}}[R(\boldsymbol{x}, y, \boldsymbol{z})] \right], \quad (1)$$

where the dataset $\mathcal{D}$ consists of pairs of images $\boldsymbol{x}$ and their corresponding ground-truth labels $y$. In practice, we employ Proximal Policy Optimization (PPO) (Schulman et al., 2017) to maximize this objective, which is equivalent to minimizing a surrogate loss $\mathcal{L}_{PPO}$.

To provide fine-grained supervision, we construct the reward $R$ by combining classification accuracy and the probability margin (Deng et al., 2022; Kwon et al., 2024):

$$R(\boldsymbol{x}, y, \boldsymbol{z}) = \lambda_{\text{acc}} \cdot \mathbb{I}(\widehat{y} = y) \quad (2)$$
$$+ \lambda_{\text{margin}} \cdot \left( p(y|\boldsymbol{x}, \boldsymbol{z}) - \max_{j \neq y} p(j|\boldsymbol{x}, \boldsymbol{z}) \right),$$

where $\lambda_{\text{acc}}$ and $\lambda_{\text{margin}}$ are weighting coefficients, $\widehat{y}$ is the predicted class, $\mathbb{I}(\cdot)$ is the indicator function, and the second term measures the confidence gap between the ground truth and the top incorrect class.

## 4. Method

The overall framework of our CRL-BPT method is illustrated in Fig. 2. Building on the formulation defined in Sec. 3, we propose two key components to regularize the optimization process: a **Curriculum Learning Strategy** and **Training Stabilization Mechanisms**.

## 4.1. Curriculum Learning Strategy

To mitigate the cold start problem (Deng et al., 2022), we introduce a curriculum learning strategy that regularizes the agent's exploration. This strategy guides the policy $\pi_\theta$ to evolve from imitating high-quality reference prompts to innovating novel prompt patterns.

**Imitation Loss.** To minimize blind exploration and mitigate the cold start problem, we introduce an imitation loss

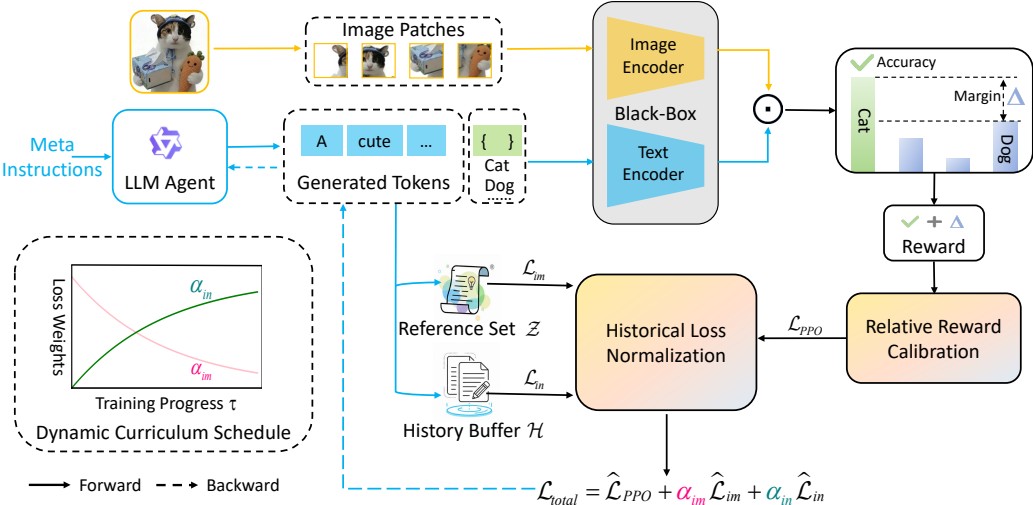

*Figure 2.* Overview of the CRL-BPT framework. CRL-BPT employs an LLM to generate interpretable prompts and guides exploration using the curriculum scheduled imitation and innovation losses. To stabilize training, CRL-BPT further incorporates historical loss normalization to align training objective scales and relative reward calibration to reduce reward variance.

$\mathcal{L}_{\text{im}}$ that guides the agent to mimic patterns from high-quality reference prompts. Specifically, we construct a reference set $\mathcal{Z}$ containing manually crafted prompt templates (see Appendix C.3 for examples), and formulate $\mathcal{L}_{\text{im}}$ as a similarity-weighted negative log-likelihood:

$$\mathcal{L}_{\text{im}} = -\mathbb{E}_{\boldsymbol{z} \sim \pi_\theta(\cdot|\boldsymbol{s})} \left[ \max_{\boldsymbol{z}' \in \mathcal{Z}} \phi(\boldsymbol{z}, \boldsymbol{z}') \cdot \log \pi_\theta(\boldsymbol{z}|\boldsymbol{s}) \right], \quad (3)$$

where $\phi(\cdot, \cdot)$ denotes the ROUGE-L (Lin, 2004) metric, which measures the longest common subsequence between the generated and reference prompts. By minimizing $\mathcal{L}_{\text{im}}$, we explicitly encourage the agent to preserve critical linguistic structures (e.g., key phrases like "a photo of a") found in the reference set, effectively injecting prior knowledge into the policy.

**Innovation Loss.** To prevent the agent from getting stuck in local optima or overfitting to the reference prompts, we introduce an innovation loss $\mathcal{L}_{\text{in}}$. We maintain a prompt history buffer $\mathcal{H}$ containing the $K$ most recently generated prompts. The novelty score of a current prompt $\boldsymbol{z}$ is measured by its dissimilarity to $\mathcal{H}$, defined as $w(\boldsymbol{z}) = 1 - \max_{\boldsymbol{z}' \in \mathcal{H}} \phi(\boldsymbol{z}, \boldsymbol{z}')$. Consequently, $\mathcal{L}_{\text{in}}$ is defined to encourage the generation of prompts with high novelty scores:

$$\mathcal{L}_{\text{in}} = -\mathbb{E}_{\boldsymbol{z} \sim \pi_\theta(\cdot|\boldsymbol{s})} \left[ w(\boldsymbol{z}) \cdot \log \pi_\theta(\boldsymbol{z}|\boldsymbol{s}) \right]. \quad (4)$$

By minimizing $\mathcal{L}_{\text{in}}$, the agent is incentivized to explore the prompt space by deviating from previously visited regions, thereby maintaining diversity in the optimization process.

**Dynamic Curriculum Schedule.** To facilitate a smooth transition from imitation to innovation, we design a dynamic

curriculum schedule. Our goal is to construct a weighting scheme that prioritizes prior knowledge in the early stages while progressively encouraging exploration once the agent acquires basic patterns. To achieve this, we employ an exponential decay mechanism. Let $\tau \in [0, 1]$ denote the training progress (i.e., current step divided by total steps). We define two dynamic weighting coefficients, $\alpha_{\text{im}}(\tau)$ and $\alpha_{\text{in}}(\tau)$, which evolve as follows:

$$\alpha_{\text{im}}(\tau) = e^{-\gamma\tau}, \quad \alpha_{\text{in}}(\tau) = 1 - e^{-\gamma\tau}, \quad (5)$$

where $\gamma$ is a hyperparameter controlling the annealing rate. As shown in Fig. 2, this schedule ensures distinct behaviors across training stages: initially (when $\tau \approx 0$), $\alpha_{\text{im}}$ dominates to enforce imitation of reference prompts; as training progresses (as $\tau \to 1$), $\alpha_{\text{im}}$ decays smoothly while $\alpha_{\text{in}}$ increases, shifting the optimization focus towards discovering novel patterns.

Finally, we integrate these curriculum objectives with the standard PPO optimization. The total loss function $\mathcal{L}$ to be minimized is formulated as:

$$\mathcal{L} = \mathcal{L}_{\text{PPO}} + \alpha_{\text{im}}(\tau) \cdot \mathcal{L}_{\text{im}} + \alpha_{\text{in}}(\tau) \cdot \mathcal{L}_{\text{in}}. \quad (6)$$

### 4.2. Training Stabilization Mechanism

Direct optimization of the objective in Eq. (6) faces two primary challenges. First, the loss terms $\mathcal{L}_{\text{PPO}}$, $\mathcal{L}_{\text{im}}$, and $\mathcal{L}_{\text{in}}$ exhibit disparate magnitudes. Consequently, the term with the largest scale tends to dominate the gradient direction, rendering the curriculum schedule ineffective. Second, raw rewards suffer from high variance due to fluctuating batch difficulty, leading to unstable policy updates. To ensure

training stability, we propose: (1) **Historical Loss Normalization** to align objective scales; and (2) **Relative Reward Calibration** to reduce reward variance.

**Historical Loss Normalization.** To ensure comparable scales across different objectives, we normalize each loss term based on its running magnitude. Specifically, for each loss term $\mathcal{L}_k \in \{\mathcal{L}_{PPO}, \mathcal{L}_{im}, \mathcal{L}_{in}\}$, we maintain an exponential moving average (EMA) of its absolute value:

$$m_k^{(t)} = \beta \cdot m_k^{(t-1)} + (1 - \beta) \cdot |\mathcal{L}_k^{(t)}|, \qquad (7)$$

where $\beta$ is a smoothing factor and $m_k$ is initialized with the first observed loss value. The normalized loss is then computed as:

$$\widehat{\mathcal{L}}_k^{(t)} = \frac{\mathcal{L}_k^{(t)}}{m_k^{(t)}}. \qquad (8)$$

This scaling aligns the magnitudes of all loss terms, ensuring that the optimization priorities are effectively controlled by the curriculum coefficients $\alpha(\tau)$. The final objective is:

$$\mathcal{L}_{total} = \widehat{\mathcal{L}}_{PPO} + \alpha_{im}(\tau)\widehat{\mathcal{L}}_{im} + \alpha_{in}(\tau)\widehat{\mathcal{L}}_{in}. \qquad (9)$$

**Relative Reward Calibration.** To mitigate the high variance caused by stochastic mini-batches, we propose to calibrate the reward signal relative to a stable baseline anchor. Specifically, we employ a fixed baseline prompt $z_{base}$ (i.e., "a photo of a {}.") and pre-compute its global expected reward $\mu_{base} = \mathbb{E}[r_{base}]$ over the training set.

For each mini-batch, we compute both the generated prompt's reward $r_\pi$ and the baseline reward $r_{base}$. The batch bias is first estimated as:

$$b_t = r_{base} - \mu_{base}, \qquad (10)$$

which indicates whether the current batch is harder ($b_t < 0$) or easier ($b_t > 0$) than average. Then, we compute the calibrated reward by subtracting this bias:

$$R = r_\pi - \eta \cdot b_t, \qquad (11)$$

where $\eta$ is a control coefficient that scales the batch bias. Intuitively, if a batch is unusually easy (i.e., high $b_t$), we discount $r_\pi$ accordingly, and vice versa. Crucially, since $\mathbb{E}[b_t] = 0$, this formulation guarantees that the calibrated reward remains *unbiased* (i.e., $\mathbb{E}[R] = \mathbb{E}[r_\pi]$) regardless of the choice of $\eta$.

Our goal is to find the optimal coefficient $\eta^*$ that minimizes the variance of $R$. As proved in Appendix A, $\eta^*$ is derived as the ratio of covariance to variance:

$$\eta^* = \frac{\text{Cov}(r_\pi, r_{base})}{\text{Var}(r_{base})}. \qquad (12)$$

Furthermore, the magnitude of variance reduction depends on the correlation between $r_\pi$ and $r_{base}$. In our setting, since both are evaluated on identical mini-batches, they exhibit high correlation, leading to substantial variance reduction.

In practice, we estimate the required statistics using EMA:

$$\begin{aligned}
\widehat{\sigma}_{base}^2 &\leftarrow \beta \cdot \widehat{\sigma}_{base}^2 + (1 - \beta) \cdot b_t^2, \\
\widehat{\sigma}_{cross}^2 &\leftarrow \beta \cdot \widehat{\sigma}_{cross}^2 + (1 - \beta) \cdot (r_\pi - \bar{r}_\pi) \cdot b_t,
\end{aligned} \qquad (13)$$

where $\widehat{\sigma}_{base}^2$ estimates the variance of $r_{base}$, $\widehat{\sigma}_{cross}^2$ tracks the covariance between $r_\pi$ and $r_{base}$, and $\bar{r}_\pi$ is the moving average of $r_\pi$. The coefficient is applied as $\eta^* = \widehat{\sigma}_{cross}^2 / \widehat{\sigma}_{base}^2$.

## 5. Experiments

### 5.1. Experimental Setup

**Datasets.** We conduct experiments across 13 datasets spanning diverse visual domains: (1) **General Object Classification**: ImageNet (Deng et al., 2009) and Caltech101 (Fei-Fei et al., 2004); (2) **Fine-Grained Classification**: OxfordPets (Parkhi et al., 2012), Flowers102 (Nilsback & Zisserman, 2008), Food101 (Bossard et al., 2014), FGVCAircraft (Maji et al., 2013), and Stanford-Cars (Krause et al., 2013); (3) **Scene and Texture Recognition**: SUN397 (Xiao et al., 2010) and DTD (Cimpoi et al., 2014); and (4) **Specialized Tasks**: Resisc45 (remote sensing) (Cheng et al., 2017), SVHN (digits) (Netzer et al., 2011), CLEVR (visual reasoning) (Johnson et al., 2017), and UCF101 (action recognition) (Soomro et al., 2012).

Furthermore, to assess robustness against distribution shifts, we evaluate on four Out-Of-Distribution (OOD) variants of ImageNet: ImageNetV2 (Recht et al., 2019), ImageNet-Sketch (Wang et al., 2019), ImageNet-A (Hendrycks et al., 2021b), and ImageNet-R (Hendrycks et al., 2021a). Detailed dataset statistics are provided in Appendix C.1.

**Evaluation Protocols and Query Budget.** We follow the standard evaluation protocols established by CoOp (Zhou et al., 2022), assessing classification accuracy under four settings: few-shot learning, base-to-new generalization, cross-dataset transfer, and out-of-distribution generalization. To rigorously test query efficiency, we impose a significantly stricter budget of **2,000 API calls**, compared to the 5,000 calls typically allocated in prior works (Park et al., 2025).

**Baselines.** We compare CRL-BPT against a series of black-box baselines, categorized from three groups: (1) **Hand-crafted Prompts**: Zero-shot CLIP with manual prompt templates. (2) **Soft Prompt Tuning**: Derivative-free approaches including BAR (Tsai et al., 2020), Black-VIP (Oh et al., 2023), BPTVLM (Yu et al., 2023), and ZIP (Park et al., 2025). (3) **Hard Prompt Tuning**: We

*Table 1.* Few-shot performance on 13 datasets. All results are reported under the 16-shot setting. **Bold** indicates the best performance, and underlined values denote the second best.

| Method | Caltech | Pets | Flowers | Food | Aircraft | SUN | DTD | SVHN | Cars | Resisc | CLEVR | UCF | ImageNet | Average |
|---|---|---|---|---|---|---|---|---|---|---|---|---|---|---|
| Zero-shot CLIP | 93.2 | 89.1 | 70.8 | 85.9 | 24.8 | 62.6 | 44.1 | 19.2 | 65.6 | 57.2 | 15.2 | 67.5 | 66.7 | 58.6 |
| BAR | 92.3 | 88.2 | 70.4 | 83.9 | 22.1 | 61.6 | 43.6 | 26.5 | 61.4 | 57.1 | 23.1 | 65.2 | 64.0 | 58.4 |
| BlackVIP | 92.4 | 87.7 | 68.4 | 82.8 | 22.2 | 61.2 | 41.6 | 22.7 | 61.8 | 55.0 | 25.2 | 64.8 | 63.5 | 57.7 |
| BPTVLM | 88.9 | 89.3 | 65.4 | 83.9 | 23.1 | 54.0 | 42.5 | 32.4 | 62.0 | 57.3 | 25.2 | 63.2 | 58.1 | 57.3 |
| ZIP | 92.1 | 91.2 | 67.5 | 86.0 | 25.3 | 60.7 | 45.7 | 44.4 | 65.7 | **63.1** | 21.2 | 68.2 | 64.1 | 61.2 |
| StablePrompt | 92.7 | 87.8 | 71.0 | 84.4 | 22.9 | 62.7 | 43.6 | 35.7 | 64.5 | 57.8 | 27.2 | 65.9 | 66.7 | 60.2 |
| CRL-BPT (Ours) | **94.6** | **92.1** | **74.1** | **86.4** | **25.6** | **66.2** | **47.0** | **46.8** | **66.2** | 62.4 | **32.3** | **70.4** | **67.8** | **64.0** |

*Table 2.* Base-to-new generalization performance.

| Method | Set | Caltech | Pets | Flowers | Food | Aircraft | SUN | DTD | SVHN | Cars | Resisc | CLEVR | UCF | ImageNet | Average |
|---|---|---|---|---|---|---|---|---|---|---|---|---|---|---|---|
| BAR | | 96.4 | 90.8 | 71.3 | 88.3 | 25.4 | 68.5 | 54.3 | 33.5 | 59.5 | 71.6 | 35.0 | 68.7 | 69.7 | 64.1 |
| BlackVIP | | 96.2 | 90.1 | 70.6 | 87.3 | 25.4 | 68.3 | 51.4 | 29.5 | 59.6 | 69.1 | 49.8 | 70.1 | 68.9 | 64.3 |
| BPTVLM | Base | 92.5 | 93.7 | 65.9 | 89.0 | 27.4 | 62.4 | 54.5 | 46.6 | 59.5 | 75.2 | 46.0 | 70.4 | 66.6 | 65.4 |
| ZIP | | 95.5 | **94.5** | 69.7 | **89.8** | 30.3 | 68.4 | **58.7** | 53.4 | **64.0** | 80.5 | 38.7 | **73.9** | 71.3 | 68.4 |
| StablePrompt | | 95.7 | 92.1 | 68.3 | 88.4 | 28.8 | 69.8 | 55.9 | 49.2 | 62.2 | 72.4 | 55.4 | 69.2 | 72.5 | 67.7 |
| CRL-BPT (Ours) | | **97.6** | 93.3 | **76.0** | 89.5 | **30.4** | **72.7** | 57.3 | 62.3 | 63.9 | 79.7 | **61.5** | 73.2 | **73.5** | **71.6** |
| BAR | | **94.3** | 95.3 | 77.8 | 89.6 | 31.2 | 73.9 | 54.9 | 29.1 | 72.5 | 62.3 | 27.1 | 76.2 | 65.1 | 65.3 |
| BlackVIP | | 92.8 | 94.6 | 75.6 | 88.9 | 32.4 | 74.3 | **58.3** | 31.4 | 71.5 | 59.9 | 31.5 | 74.6 | 65.9 | 65.5 |
| BPTVLM | New | 92.7 | 93.0 | 72.9 | 89.4 | 31.7 | 61.9 | 51.4 | 40.0 | 71.0 | 53.9 | 26.0 | 68.3 | 54.4 | 62.0 |
| ZIP | | 93.3 | 96.2 | 72.1 | 89.7 | 30.2 | 70.6 | 50.4 | 42.2 | 72.8 | 63.1 | 26.3 | 69.6 | 65.1 | 64.7 |
| StablePrompt | | 93.3 | 95.4 | 76.8 | 90.1 | 33.3 | 72.8 | 56.5 | 47.6 | 73.3 | 65.9 | 32.1 | 73.3 | 66.9 | 67.5 |
| CRL-BPT (Ours) | | 93.8 | **97.8** | **78.9** | **90.5** | **35.1** | **76.2** | 56.8 | 52.7 | **73.8** | 65.9 | **35.8** | **77.7** | **68.6** | **69.5** |
| BAR | | 95.3 | 93.0 | 74.4 | 89.0 | 28.0 | 71.1 | 54.6 | 31.1 | 65.4 | 66.7 | 30.6 | 72.3 | 67.3 | 64.5 |
| BlackVIP | | 94.5 | 92.3 | 73.0 | 88.1 | 28.5 | 71.1 | 54.6 | 30.4 | 65.0 | 64.2 | 38.6 | 72.3 | 67.4 | 64.6 |
| BPTVLM | Harmonic | 92.6 | 93.3 | 69.2 | 89.2 | 29.4 | 62.1 | 52.9 | 43.0 | 64.8 | 62.8 | 33.2 | 69.4 | 59.9 | 63.2 |
| ZIP | | 94.4 | 95.4 | 70.9 | 89.7 | 30.3 | 69.5 | 54.2 | 47.1 | 68.2 | 70.7 | 31.3 | 71.7 | 68.0 | 66.3 |
| StablePrompt | | 94.5 | 93.7 | 72.3 | 89.2 | 30.9 | 71.2 | 56.2 | 48.4 | 67.3 | 69.0 | 40.6 | 71.2 | 69.6 | 67.5 |
| CRL-BPT (Ours) | | **95.7** | **95.5** | **77.4** | **90.0** | **32.6** | **74.4** | **57.0** | **57.1** | **68.5** | **72.1** | **45.3** | **75.3** | **71.0** | **70.1** |

reproduce StablePrompt (Kwon et al., 2024), the leading RL-based approach for generating interpretable prompts.

**Implementation Details.** We adopt the pre-trained CLIP model with ViT-B/16 as the black-box environment. For the policy agent, we utilize the open-sourced Qwen2.5-3B-Instruct (Yang et al., 2024) LLM. The agent is fine-tuned via LoRA (Hu et al., 2022) with a rank of 16, a scaling factor of 32, and a dropout rate of 0.05. Optimization is performed using PPO (Schulman et al., 2017) with a learning rate of $5 \times 10^{-5}$ and a batch size of 32. Full hyperparameter settings and detailed configurations are provided in Appendix C.2. Code is available at `https://github.com/GongShuai8210/CRL-BPT`.

### 5.2. Overall Performance

**Few-Shot Performance.** Table 1 presents the 16-shot classification results across 13 datasets. Under the strict budget of 2,000 queries, CRL-BPT achieves the highest average accuracy of 64.0% and ranks first on 12 out of 13 datasets. Specifically, CRL-BPT surpasses the SOTA soft prompt method, ZIP, by 2.8%, and the leading hard prompt method, StablePrompt, by 3.8%. Furthermore, it demonstrates substantial improvements over earlier BBPT approaches, outperforming BAR by 5.6%, BlackVIP by 6.3%, and BPTVLM by 6.7%

**Base-to-New Generalization.** Table 2 reports the base-to-new generalization performance, evaluating the model's ability to generalize from seen (base) to unseen (new) classes.

*Table 3.* Cross-dataset transfer and out-of-distribution generalization performance.

| Method | Source | CDT Target | | | | | | | | | | | | | OOD Target | | | | |
|---|---|---|---|---|---|---|---|---|---|---|---|---|---|---|---|---|---|---|---|
| | ImageNet | Caltech | Pets | Flowers | Food | Aircraft | SUN | DTD | SVHN | Cars | Resisc | CLEVR | UCF | Average | ImageNet-A | ImageNetV2 | ImageNet-R | ImageNet-S | Average |
| BAR | 64.0 | 92.2 | **88.2** | 70.4 | 83.9 | 21.2 | 61.1 | 40.5 | 19.5 | 60.8 | 50.2 | 16.0 | 64.8 | 55.8 | 40.2 | 57.5 | 71.8 | 43.8 | 53.3 |
| BlackVIP | 63.5 | 92.5 | 87.5 | 67.7 | 81.7 | **22.4** | 61.1 | 40.5 | 14.6 | 61.7 | 53.6 | 16.7 | 64.6 | 55.4 | 37.0 | 57.0 | 71.3 | 43.0 | 52.1 |
| BPTVLM | 58.1 | 80.2 | 74.3 | 48.9 | 79.9 | 19.4 | 47.1 | 32.9 | 15.8 | 56.6 | 43.6 | 14.9 | 55.3 | 47.4 | 37.3 | 49.1 | 66.0 | 35.0 | 46.9 |
| ZIP | 64.1 | 89.9 | 84.9 | 64.3 | 82.8 | 20.3 | 57.1 | 39.8 | 21.8 | 61.5 | 52.2 | 14.9 | 60.7 | 54.2 | 46.3 | 57.9 | 74.0 | 44.4 | 55.6 |
| StablePrompt | 66.4 | 92.5 | 82.1 | 68.8 | 83.1 | 22.2 | 62.0 | **47.9** | 13.3 | 62.2 | 53.2 | 14.4 | 64.2 | 55.5 | 48.0 | 60.2 | 74.2 | 46.5 | 57.2 |
| CRL-BPT (Ours) | **67.8** | **93.2** | 85.9 | **70.8** | **85.0** | 21.4 | **64.5** | 44.3 | **24.1** | **63.6** | **54.1** | 12.8 | **67.1** | **57.2** | 49.6 | 61.6 | 75.3 | 47.1 | **58.4** |

*Table 4.* Representative interpretable prompts discovered by CRL-BPT. **Bold** text highlights key semantic descriptors.

| Dataset | Discovered Prompt ({} is class label) |
|---|---|
| ImageNet | A **well-composed** photo showcasing an unusual yet intriguing instance of a {}. |
| Aircraft | A captivating photo showcasing a stunning {} **aircraft model**. |
| DTD | A clear picture showing **intricate details** of a {}. |
| UCF | An image demonstrating {} **performed or exhibited** according to its standard definition. |
| SVHN | A **close-up** photograph of the {}. |

*Table 5.* Ablation study on auxiliary losses and stabilization mechanisms under the 16-shot setting. **HLN**: Historical Loss Normalization; **RRC**: Relative Reward Calibration.

| Method | Caltech | Flowers | UCF | Average |
|---|---|---|---|---|
| CRL-BPT | **94.6** | **74.1** | **70.4** | **79.7** |
| *Impact of Losses* | | | | |
| w/o $\mathcal{L}_{im}$ | 93.2 | 72.2 | 67.5 | 77.6 |
| w/o $\mathcal{L}_{in}$ | 93.8 | 72.4 | 67.7 | 78.0 |
| w/o $\mathcal{L}_{im}$ & $\mathcal{L}_{in}$ | 92.6 | 70.8 | 65.1 | 76.2 |
| *Impact of Stabilization* | | | | |
| w/o HLN | 93.3 | 70.9 | 67.1 | 77.4 |
| w/o RRC | 92.9 | 72.2 | 68.5 | 77.8 |
| w/o HLN & RRC | 92.7 | 71.2 | 66.3 | 76.6 |

CRL-BPT consistently dominates across all three metrics. On base classes, it attains 71.6% average accuracy, surpassing ZIP by 3.2% and StablePrompt by 3.9%; on new classes, CRL-BPT achieves 69.5%, outperforming StablePrompt by 2.0% and ZIP by 4.8%. This balanced superiority is confirmed by the harmonic mean, where CRL-BPT reaches 70.1%, exceeding StablePrompt by 2.6% and ZIP by 3.8%.

**Cross-Dataset Transfer & OOD.** Table 3 presents the results for Cross-Dataset Transfer (CDT) and OOD generalization. In the CDT setting, where models are trained on ImageNet and evaluated on 12 target datasets without adaptation, CRL-BPT secures the top position on 7 datasets, achieving a SOTA average accuracy of 57.2%. This outperforms the strongest baseline, StablePrompt, by 1.7%. Similarly, regarding OOD generalization, CRL-BPT demonstrates strong robustness across all four ImageNet variants, obtaining the best average accuracy of 58.4% (+1.2% over StablePrompt). These results confirm that the curriculum-guided exploration prevents overfitting to source domains, fostering the learning of generalizable semantic patterns.

Notably, we also scale the policy agent to Qwen2.5-7B-Instruct (Yang et al., 2024), observing only a marginal gain (<1%) over the 3B version. This demonstrates that CRL-BPT extracts high-quality prompts efficiently without relying on computationally expensive LLMs (details in Appendix D.6).

**Interpretability and Qualitative Analysis.** In addition to the examples illustrated in Fig. 1(a), Table 4 presents representative prompts discovered by CRL-BPT under the few-shot setting. These generated prompts capture domain-specific semantics. For instance, on texture recognition tasks like DTD, the agent explicitly emphasizes visual patterns (e.g., "intricate details"). Similarly, for action recognition in UCF101, the prompts focus on the execution of the activity (e.g., "performed or exhibited"). Notably, for fine-grained classification on FGVCAircraft, the agent learns to append the explicit category context ("aircraft model") to the class name. We provide comprehensive lists of discovered prompts across all datasets in Appendix D.1. Prompt length and other hyperparameter sensitivity analysis are provided in Appendix D.3.

### 5.3. Ablation Study

**Impact of Auxiliary Losses.** We evaluate the contribution of our curriculum objectives by progressively removing the imitation loss $\mathcal{L}_{im}$ and innovation loss $\mathcal{L}_{in}$. These ablations are conducted on three representative datasets: Caltech101 (general), Flowers102 (fine-grained), and UCF101 (action).

As shown in Table 5, the full CRL-BPT model achieves the best average accuracy of 79.7%. Removing $\mathcal{L}_{im}$ results in a 2.1% drop, confirming that supervision from reference prompts provides essential guidance to prevent blind explo-

*Table 6.* Impact of different curriculum schedules.

| Schedule Type | Caltech | Flowers | UCF | Average |
|---|---|---|---|---|
| CRL-BPT (Exponential) | **94.6** | **74.1** | **70.4** | **79.7** |
| Linear Schedule | 94.0 | 72.9 | 67.1 | 78.2 |
| Constant Weighting | 93.0 | 71.9 | 65.7 | 76.9 |

*Table 7.* Effect of reference set composition on test accuracy.

| Reference Set | Flowers | SUN | UCF | Avg. |
|---|---|---|---|---|
| Top-1 | 72.8 | 65.4 | 69.3 | 69.1 |
| Top-10 | 73.4 | 65.8 | 69.6 | 69.6 |
| Top-20 (default) | **74.1** | **66.2** | **70.4** | **70.2** |
| Top-40 | 73.4 | 65.1 | 69.9 | 69.5 |
| Bottom-20 | 72.5 | 65.7 | 68.1 | 68.7 |
| Random-20 | 73.0 | 65.5 | 67.8 | 68.8 |
| Top-20 (augmented) | 72.8 | 66.0 | 68.8 | 69.2 |

ration. Similarly, discarding $\mathcal{L}_{in}$ leads to a 1.7% decline, as the lack of diversity enforcement causes the policy to converge prematurely to suboptimal solutions. Crucially, removing both losses degrades performance by 3.5%. This highlights their complementary nature: $\mathcal{L}_{im}$ ensures stability via exploitation of prior knowledge, while $\mathcal{L}_{in}$ drives performance growth through exploration. An additional ablation on the similarity metric $\phi(\cdot, \cdot)$ used in these losses is presented in Appendix D.4.

**Impact of Stabilization Mechanisms.** We validate our stabilization mechanisms: Historical Loss Normalization (HLN) and Relative Reward Calibration (RRC). As shown in Table 5, removing HLN drops performance by 2.3%, as unscaled loss terms tend to dominate gradients, rendering the curriculum schedule ineffective. Excluding RRC results in a 1.9% decrease, confirming that batch-induced reward variance destabilizes policy updates. Further quantitative analysis of reward variance is detailed in Appendix D.5. Notably, removing both mechanisms degrades accuracy by 3.1%, proving that aligning objective scales and reducing reward variance are prerequisites for robust training.

**Impact of Curriculum Schedules.** We analyze the impact of the schedule function by comparing our exponential design in Eq. (5) against two variants: (1) a **Linear Schedule**, where $\alpha_{im}$ linearly decays ($1 \to 0$) while $\alpha_{in}$ linearly increases ($0 \to 1$); and (2) **Constant Weighting**, where both coefficients remain fixed at 1 throughout training. As shown in Table 6, replacing the exponential schedule with the linear one results in a 1.5% accuracy drop, while using constant weights leads to a more pronounced 2.8% decline. The exponential schedule is superior because it provides stronger guidance in the early stages (when $\tau \approx 0$), while ensuring a decisive shift towards innovation in the later stages (as $\tau \to 1$) to prevent the agent from over-relying on reference priors.

### 5.4. Analysis of Reference Set

We examine how the composition of the reference set affects performance by varying both its size and quality. As shown in Table 7, performance degrades gracefully with fewer or lower-quality reference prompts. Importantly, even with Bottom-20 or Random-20 references, CRL-BPT still outperforms StablePrompt on the same three datasets by over 2 points on average. Thus, CRL-BPT benefits from better

references, but is not highly sensitive to the exact choice of the reference set. We also tested augmenting the reference set with high-performing prompts obtained after one optimization round. Interestingly, augmentation does not improve performance; instead, the average accuracy drops from 70.2% to 69.2%. This suggests that the reference set mainly acts as a warm-start prior for imitation, rather than a pool that should be iteratively refined with self-generated prompts. A plausible explanation is that overly strong or highly similar references can bias the policy toward a narrow region of the prompt space during imitation, thereby reducing the benefit of subsequent innovation.

### 5.5. Query Efficiency

**Acceleration via Curriculum Objectives.** We first examine how our auxiliary losses impact convergence speed. Fig. 3 illustrates the training trajectories on Flowers102. The full CRL-BPT (blue line) rapidly stabilizes around 74.0% accuracy within just 800 queries. In contrast, the variant removing both imitation ($\mathcal{L}_{im}$) and innovation ($\mathcal{L}_{in}$) losses (orange line) progresses slowly, reaching approximately 71% accuracy even after consuming the full budget of 2,000 queries. This substantial gap confirms that the curriculum schedule serves as a critical accelerator, effectively guiding the agent through the vast search space.

**Optimization vs. In-Context Learning (ICL).** A common alternative to address the cold start problem in LLM-based optimization is In-Context Learning (ICL), where high-quality exemplars are provided in the input context to guide generation. To compare with this strategy, we implement an ICL-based variant, where prompts from the reference set $\mathcal{Z}$ serve as few-shot demonstrations for the LLM, rather than being used as optimization targets via $\mathcal{L}_{im}$.

As shown in Fig. 3, the ICL variant (pink line) achieves a high initial accuracy of approximately 55% thanks to the explicit guidance from exemplars. However, its performance curve remains nearly flat throughout the process, only marginally improving to around 72% after 2,000 queries. In contrast, although CRL-BPT starts with lower performance, it rapidly surpasses the ICL variant within just 600 queries and eventually converges to a much higher accuracy. This

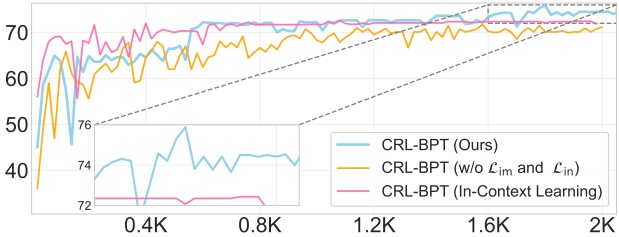

*Figure 3.* Convergence analysis comparing CRL-BPT with ablation variants and in-context learning.

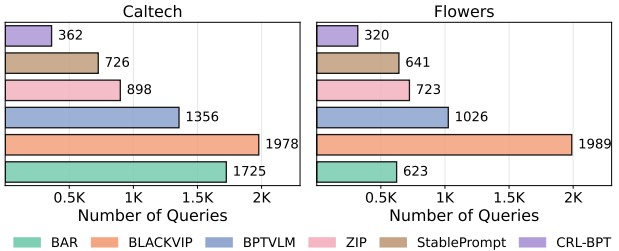

*Figure 4.* Number of API queries to reach target accuracy.

plateau in ICL likely stems from the persistent presence of exemplars, which imposes a permanent constraint confining the agent to prior patterns. Conversely, CRL-BPT employs a decaying imitation schedule to prevent the agent from over-relying on initial priors, allowing the innovation objective to discover novel prompts that outperform manual references.

**Efficiency vs. SOTA Baselines.** Fig. 1(b) illustrates that CRL-BPT consistently maintains a performance lead throughout the entire training trajectory. Furthermore, we quantify practical efficiency by measuring the number of API calls required to reach a target accuracy (defined as the minimum peak performance among all methods) (Park et al., 2025). As shown in Fig. 4, CRL-BPT exhibits superior efficiency, requiring significantly fewer queries to match this target compared to the other methods. Overall, these results confirm that CRL-BPT not only achieves higher final accuracy but also converges faster, making it an ideal solution for budget-constrained scenarios.

## 6. Conclusion

In this paper, we addressed the critical trade-off between prompt interpretability and query efficiency in black-box prompt tuning. We proposed CRL-BPT, which leverages a curriculum learning strategy to guide an LLM agent from imitating prior knowledge to discovering novel patterns, effectively overcoming the cold start problem. Coupled with our tailored training stabilization mechanisms, this approach promotes more stable optimization under stochastic black-box feedback. Extensive experiments across 13 datasets validate that CRL-BPT establishes new state-of-the-art per-

formance. Crucially, it demonstrates superior convergence speed, requiring only 2,000 API calls to generate highly interpretable, domain-specific prompts, making it a practical solution for budget-constrained real-world applications.

## Acknowledgements

This work was supported by the Taishan Scholar Program of Shandong Province (Grant No. tsqn202211199 and Grant No. tsqn202507240), the National Natural Science Foundation of China (Grant No. 62576193), and the Natural Science Foundation of Shandong Province (Grant No. ZR2025MS985).

## Impact Statement

This paper presents work whose goal is to advance the field of Machine Learning. There are many potential societal consequences of our work, none which we feel must be specifically highlighted here.

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

# A. Theoretical Analysis of Relative Reward Calibration

In this appendix, we provide the theoretical derivation of the optimal control coefficient $\eta^*$ introduced in Section 4.2.

## A.1. Problem Formulation

To mitigate the high variance of reward signals caused by stochastic mini-batches, we introduce a baseline $r_{\text{base}}$ and define the calibrated reward as:

$$R = r_\pi - \eta \cdot (r_{\text{base}} - \mu_{\text{base}}), \tag{14}$$

where $r_\pi$ and $r_{\text{base}}$ denote the rewards of the generated prompt and the fixed baseline prompt on the same batch, respectively. $\mu_{\text{base}} = \mathbb{E}[r_{\text{base}}]$ is the pre-computed expected reward. Since $\mathbb{E}[r_{\text{base}} - \mu_{\text{base}}] = 0$, we have $\mathbb{E}[R] = \mathbb{E}[r_\pi]$ for any $\eta$, ensuring that the calibrated reward remains *unbiased*. Our goal is to find the optimal $\eta^*$ that minimizes the variance of $R$.

## A.2. Derivation of Optimal Coefficient

We compute the variance of the calibrated reward $R$:

$$\text{Var}(R) = \text{Var}\left(r_\pi - \eta \cdot (r_{\text{base}} - \mu_{\text{base}})\right) \tag{15}$$
$$= \text{Var}(r_\pi - \eta \cdot r_{\text{base}}) \quad \text{(as } \mu_{\text{base}} \text{ is a constant)}$$
$$= \text{Var}(r_\pi) + \eta^2 \text{Var}(r_{\text{base}}) - 2\eta \, \text{Cov}(r_\pi, r_{\text{base}}).$$

Since variance is non-negative, the coefficient of the quadratic term, $\text{Var}(r_{\text{base}})$, is strictly positive. Thus, Eq. (15) describes a convex parabola opening upward, ensuring that the critical point is a unique global minimum.

Taking the derivative with respect to $\eta$ and setting it to zero:

$$\frac{d \, \text{Var}(R)}{d\eta} = 2\eta \, \text{Var}(r_{\text{base}}) - 2 \, \text{Cov}(r_\pi, r_{\text{base}}) = 0. \tag{16}$$

Solving for $\eta$, we obtain the optimal coefficient:

$$\eta^* = \frac{\text{Cov}(r_\pi, r_{\text{base}})}{\text{Var}(r_{\text{base}})}. \tag{17}$$

## A.3. Variance Reduction Guarantee

Substituting $\eta^*$ back into Eq. (15), we obtain the minimum variance:

$$\text{Var}(R)_{\min} = \text{Var}(r_\pi) - \frac{[\text{Cov}(r_\pi, r_{\text{base}})]^2}{\text{Var}(r_{\text{base}})} \tag{18}$$
$$= \text{Var}(r_\pi) \cdot (1 - \rho^2),$$

where

$$\rho = \frac{\text{Cov}(r_\pi, r_{\text{base}})}{\sqrt{\text{Var}(r_\pi) \cdot \text{Var}(r_{\text{base}})}} \tag{19}$$

is the Pearson correlation coefficient between $r_\pi$ and $r_{\text{base}}$. Eq. (19) provides a theoretical guarantee for our method:

---

**Algorithm 1** Curriculum Reinforcement Learning for Black-Box Prompt Tuning

---

**Require:** Dataset $\mathcal{D}$, Black-box VLM $\mathcal{M}$, LLM Policy $\pi_\theta$, Reference Set $\mathcal{Z}$, Baseline Prompt $z_{\text{base}}$, Total Steps $T$ and Meta Instructions $s$.

1: **Pre-compute:** $\mu_{\text{base}} \leftarrow \mathbb{E}_{\mathcal{D}}[r_{\text{base}}]$ for $z_{\text{base}}$
2: **for** step $t = 1$ to $T$ **do**
3:     /* Curriculum Schedule */
4:     Calculate progress $\tau \leftarrow t/T$
5:     Update weights: $\alpha_{\text{im}} \leftarrow e^{-\gamma\tau}$, $\alpha_{\text{in}} \leftarrow 1 - e^{-\gamma\tau}$
6:     /* Sampling and Evaluation */
7:     Sample batch $d \sim \mathcal{D}$
8:     Generate prompts $z \sim \pi_\theta(\cdot|s)$
9:     Query $\mathcal{M}$ to get rewards $r_\pi$ and $r_{\text{base}}$
10:    /* Relative Reward Calibration */
11:    Estimate $\eta^* = \text{Cov}(r_\pi, r_{\text{base}})/\text{Var}(r_{\text{base}})$
12:    Compute $R \leftarrow r_\pi - \eta^* \cdot (r_{\text{base}} - \mu_{\text{base}})$
13:    /* Loss Computation and Normalization */
14:    Compute raw losses: $\mathcal{L}_{\text{PPO}}, \mathcal{L}_{\text{im}}, \mathcal{L}_{\text{in}}$
15:    **for** $k \in \{\text{PPO}, \text{im}, \text{in}\}$ **do**
16:       Update scale: $m_k \leftarrow \beta \cdot m_k + (1 - \beta) \cdot |\mathcal{L}_k|$
17:       Normalize: $\widehat{\mathcal{L}}_k \leftarrow \mathcal{L}_k/m_k$
18:    **end for**
19:    /* Policy Update */
20:    Compute $\mathcal{L}_{\text{total}} \leftarrow \hat{\mathcal{L}}_{\text{PPO}} + \alpha_{\text{im}}\hat{\mathcal{L}}_{\text{im}} + \alpha_{\text{in}}\hat{\mathcal{L}}_{\text{in}}$
21:    Update $\theta \leftarrow \theta - \zeta\nabla_\theta\mathcal{L}_{\text{total}}$
22: **end for**

---

- Since $0 \leq \rho^2 \leq 1$, the term $(1 - \rho^2)$ acts as a reduction factor. This guarantees that the calibrated variance is always less than or equal to the original variance (i.e., $\text{Var}(R)_{\min} \leq \text{Var}(r_\pi)$).

- The magnitude of variance reduction depends on the correlation $|\rho|$. In our setting, since $r_\pi$ and $r_{\text{base}}$ are evaluated on *identical* image batches, they are both affected by the same batch difficulty. This dependency results in a high correlation coefficient $|\rho|$, which directly leads to substantial variance reduction.

# B. Algorithm

To facilitate reproducibility and provide a clear overview of our framework, we present the complete training process of CRL-BPT in Algorithm 1.

# C. Experimental Setup Details

## C.1. Dataset Statistics

We evaluate CRL-BPT on a diverse suite of 13 image classification datasets, covering general objects, fine-grained categories, scenes, textures, and specialized tasks (e.g., satellite imagery and visual reasoning). Additionally, to assess

*Table 8.* Statistics of the 13 standard datasets and 4 OOD benchmarks used in our experiments. **Val Size** denotes the size of the official validation split; "—" indicates that the dataset does not have an official validation split (or uses the training set of ImageNet).

| Dataset | Category | # Classes | Train Size | Val Size | Test Size |
|---|---|---|---|---|---|
| *General Object Classification* | | | | | |
| ImageNet (Deng et al., 2009) | General Objects | 1,000 | 1,281,167 | — | 50,000 |
| Caltech101 (Fei-Fei et al., 2004) | General Objects | 100 | 4,128 | 1,649 | 2,465 |
| *Fine-Grained Visual Classification* | | | | | |
| OxfordPets (Parkhi et al., 2012) | Animals | 37 | 2,944 | 736 | 3,669 |
| StanfordCars (Krause et al., 2013) | Cars | 196 | 6,509 | 1,635 | 8,041 |
| Flowers102 (Nilsback & Zisserman, 2008) | Plants | 102 | 4,093 | 1,633 | 2,463 |
| Food101 (Bossard et al., 2014) | Food | 101 | 50,500 | 20,200 | 30,300 |
| FGVCAircraft (Maji et al., 2013) | Aircrafts | 100 | 3,334 | 3,333 | 3,333 |
| *Scene, Texture, and Others* | | | | | |
| SUN397 (Xiao et al., 2010) | Scene Recognition | 397 | 15,880 | 3,970 | 19,850 |
| DTD (Cimpoi et al., 2014) | Textures | 47 | 2,820 | 1,128 | 1,692 |
| Resisc45 (Cheng et al., 2017) | Remote Sensing | 45 | 15,750 | 6,300 | 9,450 |
| SVHN (Netzer et al., 2011) | Street View Numbers | 10 | 51,280 | 21,977 | 26,032 |
| UCF101 (Soomro et al., 2012) | Action Recognition | 101 | 7,639 | 1,898 | 3,783 |
| CLEVR (Johnson et al., 2017) | Visual Reasoning | 8 | 55,999 | 14,001 | 15,000 |
| *Robustness OOD Benchmarks* | | | | | |
| ImageNetV2 (Recht et al., 2019) | Robustness | 1,000 | — | — | 10,000 |
| ImageNet-Sketch (Wang et al., 2019) | Sketch | 1,000 | — | — | 50,889 |
| ImageNet-A (Hendrycks et al., 2021b) | Adversarial | 200 | — | — | 7,500 |
| ImageNet-R (Hendrycks et al., 2021a) | Rendition | 200 | — | — | 30,000 |

robust generalization, we employ 4 Out-Of-Distribution (OOD) variants of ImageNet. Detailed statistics for all benchmarks, including the number of classes and sample splits, are summarized in Table 8.

## C.2. Implementation and Hyperparameters

**Model Architecture.** Consistent with prior works (Oh et al., 2023; Park et al., 2025), we employ the pre-trained CLIP model with a ViT-B/16 backbone as the black-box environment. For the policy agent, we utilize the open-sourced Qwen2.5-3B-Instruct (Yang et al., 2024) LLM.

**Training Settings.** To ensure efficiency, the LLM agent is fine-tuned via LoRA (Hu et al., 2022) with a rank of 16, a scaling factor of 32, and a dropout rate of 0.05. Optimization is performed using PPO (Schulman et al., 2017) with the following configurations:

- Learning rate: $5 \times 10^{-5}$
- Batch size: 32
- Clip range: 0.2
- KL penalty coefficient: 0.1
- Value function coefficient: 0.5
- Entropy bonus coefficient: 0.01
- Max gradient norm: 1.0

During sampling, we set the temperature to 0.95, top-$p$ = 1.0 (i.e., no nucleus sampling truncation), and top-$k$ = 0.

**CRL-BPT Specific Hyperparameters.** We summarize the key configurations specific to the CRL-BPT framework below. A detailed sensitivity analysis for each parameter is provided in Appendix D.3:

- **Meta instructions**: "*You are an expert in prompt engineering for Vision-Language Models. Your goal is to write a text description (prompt) that accurately classifies images.*"
- **Prompt length**: Fixed to 15 tokens (20 tokens for CLEVR due to its reasoning complexity).
- **Curriculum schedule**: The annealing rate $\gamma$ in Eq. (5) is set to 2, controlling the transition speed from imitation to innovation.
- **History buffer**: The buffer size $K$ in Eq. (4) for the innovation loss is set to 100.
- **Stabilization**: The smoothing factor $\beta$ for EMA in Eq. (7) and Eq. (13) is set to 0.99 for both loss normalization and reward calibration.
- **Reward weights**: We set $\lambda_{\text{acc}} = 30$ and $\lambda_{\text{margin}} = 10$ in Eq. (2) to balance accuracy and margin maximization.

Code is available at https://github.com/

*Table 9.* Representative manual templates from CLIP's official ensemble, used to construct $\mathcal{Z}$. "{}" denotes the class label.

| Manual Prompt Templates |
| --- |
| a photo of a {}. |
| a blurry photo of a {}. |
| a photo of the large {}. |
| a photo of the small {}. |
| a good photo of a {}. |
| a bad photo of a {}. |
| a cropped photo of the {}. |
| a close-up photo of a {}. |
| a bright photo of a {}. |
| a dark photo of a {}. |

`GongShuai8210/CRL-BPT`.

### C.3. Loss and Reward Configuration

**Imitation Loss Construction.** To construct the reference set $\mathcal{Z}$ used in Eq. (3), we leverage the 80 manually designed prompt templates[1] officially provided by CLIP (Radford et al., 2021). These templates encompass diverse linguistic patterns, such as photographic conditions and object scales. Representative examples are listed in Table 9. We evaluate each template on 200 batches of training data and select the top-20 performing prompts to form $\mathcal{Z}$. This ensures the capture of effective linguistic priors.

**Innovation and Reward Details.** For the innovation loss, the size of the history buffer $\mathcal{H}$ is set to $K = 100$ in Eq. (4). For relative reward calibration, we designate "a photo of a {}." as the baseline prompt $z_{\text{base}}$. The global expected reward $\mu_{\text{base}}$ is pre-computed using the first 200 batches.

### C.4. Baseline Details

To ensure fair comparison, we reproduce all baselines under a unified protocol with a maximum budget of 2,000 API calls.

- **Zero-shot CLIP**: We use dataset-specific templates consistent with CoOp (Zhou et al., 2022). The exact templates for all 13 datasets are detailed in Table 10.
- **Soft Prompt Tuning**: For BAR, BlackVIP, BPTVLM, and ZIP, we adopt the official implementations and hyperparameters from their original papers.
- **StablePrompt**: Originally proposed for LLMs (Kwon et al., 2024), we re-implemented and adapted this RL-based algorithm for the VLM (CLIP) setting, strictly adhering to the same query budget.

---

[1]Full list available at `https://github.com/openai/CLIP/blob/main/notebooks/Prompt_Engineering_for_ImageNet.ipynb`.

*Table 10.* List of hand-crafted prompts used for Zero-shot CLIP. "{}" denotes the class label.

| Dataset | Hand-crafted Template |
| --- | --- |
| ImageNet | a photo of a {}. |
| Caltech101 | a photo of a {}. |
| OxfordPets | a photo of a {}, a type of pet. |
| StanfordCars | a photo of a {}, a type of car. |
| Flowers102 | a photo of a {}, a type of flower. |
| Food101 | a photo of a {}, a type of food. |
| FGVCAircraft | a photo of a {}, a type of aircraft. |
| SUN397 | a photo of a {}. |
| DTD | {} texture. |
| Resisc45 | This is a photo of a {}. |
| SVHN | This is a photo of a {}. |
| UCF101 | a photo of a person doing {}. |
| CLEVR | This is a photo of {} objects. |

## D. Additional Experimental Results

### D.1. Discovered Prompts

We provide a comprehensive list of prompts discovered by CRL-BPT across all datasets. Table 11 and Table 12 detail the prompts generated under the few-shot and base-to-new settings, respectively.

Several interesting observations emerge from the discovered prompts. First, the LLM policy learns to incorporate dataset-specific descriptors that align with the visual characteristics of each domain. For instance, prompts for fine-grained datasets often include quality-related terms (e.g., "clear", "detailed", "vivid") that encourage the VLM to focus on discriminative features. Second, some prompts contain unconventional patterns such as special characters or seemingly redundant phrases, which may serve as implicit regularization or help the model attend to specific visual attributes. Third, the prompts exhibit diversity across datasets, suggesting that the curriculum-based RL framework successfully adapts the prompt style to different visual recognition tasks rather than converging to a generic template.

### D.2. Reward Coefficients $\lambda_{\text{acc}}$ and $\lambda_{\text{margin}}$.

The reward signal combines classification accuracy and softmax margin, weighted by $\lambda_{\text{acc}}$ and $\lambda_{\text{margin}}$ respectively. We conduct a grid search over both coefficients to understand their joint effect. As shown in Fig. 5, the two components exhibit complementary characteristics. When $\lambda_{\text{acc}} = 0$ (first row), relying solely on the margin signal yields suboptimal performance, as margin alone lacks direct supervision on classification correctness. Conversely, setting $\lambda_{\text{margin}} = 0$ (first column) also underperforms, since the margin signal provides fine-grained discrimination information that

*Table 11.* Examples of text prompts discovered by the LLM policy for few-shot classification. "{}" denotes the class label.

| Dataset | Discovered Prompt ({} is class label) |
|---|---|
| ImageNet | A well-composed photo showcasing an unusual yet intriguing instance of a {}. |
| Caltech101 | A detailed **painting/photorealistic illustration** of an elegant {}. |
| OxfordPets | Low-resolution good quality photo of my low-detail pet {}. |
| StanfordCars | a clear, vivid **unique** and distinctive {}. Accuracy: 7 |
| Flowers102 | Lovely sunny day, capture a beautiful {} blooming beautifully in the garden. |
| Food101 | A clear photograph of the delicious {}. How tasty! |
| FGVCAircraft | A captivating photo showcasing a stunning **{}** aircraft model. |
| SUN397 | Low-quality photograph showing clear details of an {}, with noticeable features emphasized. |
| DTD | A clear picture showing intricate details of a {}. |
| Resisc45 | A detailed image showing a {}, extracted directly from real-world imagery. |
| SVHN | A close-up photograph of the "{}". |
| UCF101 | an image demonstrating **{}** performed or exhibited according to its standard definition. |
| CLEVR | a unique photo of various interacting together. This prompt leverages several elements likely used prevalently within |

*Table 12.* Examples of text prompts discovered by the LLM policy for base-to-new generalization. "{}" denotes the class label.

| Dataset | Discovered Prompt ({} is class label) |
|---|---|
| ImageNet | a visually distinctive photo of an extraordinary {} with unique characteristics. This expression |
| Caltech101 | a clearly photographed # {}. Photo must show#{} #photographically)# photograph |
| OxfordPets | A poorly posed poor quality poorly lit poorly behaved pet {}. |
| StanfordCars | a clear photo of [the / a] unique {} with details visible. |
| Flowers102 | A vibrant and clear **close-up** view of a stunning **{}**, |
| Food101 | A clean shot of the **{}** dish. ### Explanation: - The |
| FGVCAircraft | a vibrant, well-lit photograph showcasing a distinctive **{}** aircraft model |
| SUN397 | a clear image showing a well-lit view of a typical sized {}, clearly |
| DTD | a picture of {}. This encourages creativity while keeping things approachable and encouraging imagination |
| Resisc45 | a detailed or unique perspective shot showing/images of multiple/few {}, including variations |
| SVHN | "'plaintext a difficult-to-distinguish photograph of {} "' This prompt |
| UCF101 | a clear yet slightly distorted photograph capturing an individual performing {}. This prompt aims |
| CLEVR | a detailed illustration or depiction showing multiple instances of various objects cohesively arranged in natural settings like |

pure accuracy cannot capture. The optimal performance (74.10%) is achieved at $\lambda_{acc} = 30$ and $\lambda_{margin} = 10$, indicating that accuracy should be weighted more heavily as the primary objective, while margin serves as a complementary signal to improve decision boundaries. Notably, excessively large values for either coefficient (*e.g.*, 100) lead to performance degradation, suggesting that balanced weighting is crucial for stable optimization.

### D.3. Hyperparameter Sensitivity Analysis

We conduct a comprehensive sensitivity analysis on five key hyperparameters: curriculum rate $\gamma$, prompt length $L$, history buffer size $K$, EMA factor $\beta$, and reward coefficients $\lambda_{acc}$ and $\lambda_{margin}$.

**Prompt Length $L$.** The prompt length determines the number of learnable tokens that the LLM generates for CLIP. As shown in Table 13, the optimal prompt length varies across datasets with different complexity levels. For standard image classification datasets such as Flowers and UCF, a prompt length of 15 achieves the best performance

*Table 13.* Effect of the prompt length $L$.

| $L$ | Flowers | UCF | CLEVR | Avg. |
|---|---|---|---|---|
| 5 | 72.2 | 66.9 | 22.0 | 53.7 |
| 10 | 73.4 | 68.5 | 23.1 | 55.0 |
| 15 | **74.1** | **70.4** | 26.1 | 56.9 |
| 20 | 71.4 | 70.1 | 32.3 | **57.9** |
| 25 | 71.9 | 68.7 | 30.6 | 57.0 |
| 50 | 70.1 | 67.5 | **34.0** | 57.2 |

(74.1% and 70.4%, respectively), and longer prompts do not yield further improvements. This suggests that relatively short prompts are sufficient to capture the discriminative semantics required for conventional visual recognition tasks. In contrast, for the complex visual reasoning dataset CLEVR, which requires understanding spatial relationships, object attributes, and compositional concepts, longer prompts consistently improve performance, with the best accuracy (34.0%) achieved at a prompt length of 50. This indicates that complex reasoning tasks benefit from richer prompt representations that can encode more sophisticated visual-semantic relationships.

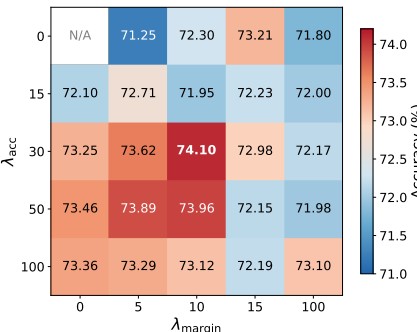

*Figure 5.* Effect of the reward coefficients $\lambda_{\text{acc}}$ and $\lambda_{\text{margin}}$. The heatmap illustrates the grid search results. "N/A" indicates an invalid configuration where both reward signals are absent.

**Curriculum Schedule $\gamma$.** The coefficient $\gamma$ controls the transition rate between imitation and innovation phases in our curriculum learning schedule (Eq. 5). A smaller $\gamma$ leads to slower transitions, allowing the model to spend more time in each phase, while a larger $\gamma$ accelerates the curriculum progression. As shown in Table 14, performance is sensitive to this hyperparameter. When $\gamma$ is too small (*e.g.*, 1.0), the slow transition extends the imitation phase, limiting the model's ability to explore novel prompt patterns. Conversely, when $\gamma$ is too large (*e.g.*, 5.0), the rapid transition causes the model to prematurely shift toward innovation before sufficiently learning from reference prompts. The optimal performance is achieved at $\gamma = 2.0$, which provides a balanced curriculum that allows adequate imitation learning in early stages while gradually encouraging exploration. This suggests that a moderate transition rate is crucial for effective curriculum-based prompt optimization.

*Table 14.* Effect of the curriculum schedule coefficient $\gamma$.

| $\gamma$ | Flowers | Food | UCF | Avg. |
|---|---|---|---|---|
| 1.0 | 72.8 | 85.5 | 67.1 | 75.1 |
| 1.5 | 73.3 | 86.0 | 68.9 | 76.1 |
| 2.0 | **74.1** | **86.4** | **70.4** | **77.0** |
| 2.5 | 72.5 | 86.2 | 68.4 | 75.7 |
| 3.0 | 72.2 | 86.2 | 67.9 | 75.4 |
| 5.0 | 71.1 | 85.9 | 67.5 | 74.8 |

**History Buffer Size $K$.** The history buffer $\mathcal{H}$ stores the $K$ most recently generated prompts for computing the innovation loss, which encourages the model to explore novel patterns by penalizing similarity to historical outputs. As shown in Table 15, the buffer size $K$ exhibits a trade-off between diversity encouragement and pattern stability. When $K$ is too small (*e.g.*, 50), the limited historical context fails to capture sufficient prompt patterns, resulting in inadequate diversity regularization. Conversely, when $K$ is too large (*e.g.*, 2000), the buffer accumulates prompts from early training stages that may no longer be representative, intro-

*Table 15.* Effect of the history buffer size $K$.

| $K$ | Flowers | Food | UCF | Avg. |
|---|---|---|---|---|
| 50 | 73.9 | 86.0 | 68.6 | 76.2 |
| 100 | 74.1 | 86.4 | 70.4 | 77.0 |
| 200 | **74.5** | 85.8 | 70.9 | **77.1** |
| 500 | 73.5 | **87.6** | 69.9 | 77.0 |
| 1000 | 71.6 | 86.8 | **71.6** | 76.7 |
| 2000 | 70.6 | 86.1 | 69.4 | 75.4 |

*Table 16.* Effect of the EMA factor $\beta$.

| $\beta$ | Flowers | Food | UCF | Avg. |
|---|---|---|---|---|
| 0.90 | 72.9 | 85.3 | 67.9 | 75.4 |
| 0.95 | 73.3 | 86.8 | 68.1 | 76.1 |
| 0.99 | **74.1** | 86.4 | **70.4** | **77.0** |
| 0.999 | 73.9 | **86.5** | 69.8 | 76.7 |

ducing noise into the diversity computation and degrading performance. The results show that moderate buffer sizes ($K \in [100, 500]$) achieve consistently strong performance, with the average accuracy ranging from 77.0% to 77.1%.

**EMA Factor $\beta$.** The EMA factor $\beta$ controls the smoothing rate for both historical loss normalization and relative reward calibration. As shown in Table 16, when $\beta$ is too small (*e.g.*, 0.9), the statistics update rapidly, causing the normalization scales and reward estimates to fluctuate excessively and resulting in unstable training dynamics. Conversely, when $\beta$ is too large (*e.g.*, 0.999), the statistics adapt too slowly to reflect the evolving loss landscape and reward distribution during training, leading to suboptimal calibration. The best performance is achieved at $\beta = 0.99$, which allows the model to maintain smooth estimates while still adapting to meaningful changes in the optimization process.

**D.4. Analysis of Similarity Metric $\phi(\cdot, \cdot)$.**

The imitation loss $\mathcal{L}_{\text{im}}$ and innovation loss $\mathcal{L}_{\text{in}}$ rely on a similarity metric $\phi(\cdot, \cdot)$ to measure the resemblance between prompts. We compare our default choice, ROUGE-L, against cosine similarity computed via the CLIP text encoder. As shown in Table 17, ROUGE-L outperforms cosine similarity by 1.5% in average accuracy. This performance gap can be attributed to the nature of the two metrics: ROUGE-L captures structural similarity by measuring the longest common subsequence between prompts, making it sensitive to shared linguistic patterns such as key phrases and syntactic structures. In contrast, cosine similarity operates on semantic embeddings, which may overlook fine-grained lexical overlaps that are critical for prompt optimization. Since our curriculum strategy explicitly aims to guide the agent toward imitating specific linguistic patterns from reference prompts and deviating from previously

*Table 17.* Ablation on similarity metric $\phi(\cdot, \cdot)$.

| Method | Flowers | Food | UCF | Avg. |
|---|---|---|---|---|
| ROUGE-L | **74.1** | **86.4** | **70.4** | **77.0** |
| Cosine Sim. | 72.9 | 84.8 | 68.9 | 75.5 |

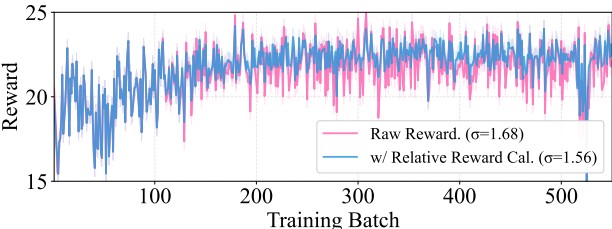

*Figure 6.* Visualization of reward signal stability on Flowers102.

generated ones, ROUGE-L provides more appropriate supervision for this objective.

### D.5. Reward Variance Analysis.

To validate the effectiveness of relative reward calibration, we visualize the reward signals across training batches on Flowers102. As shown in Fig. 6, the raw reward signal (pink) exhibits substantial fluctuations, with a standard deviation of $\sigma = 1.68$. These fluctuations arise from varying batch difficulties: some batches contain easily classifiable images while others are inherently more challenging, causing the reward to fluctuate regardless of the actual prompt quality.

By applying our relative reward calibration, the calibrated reward signal (blue) achieves a reduced standard deviation of $\sigma = 1.56$, corresponding to a 7.1% relative reduction. This improvement confirms that our mechanism successfully compensates for batch-level biases by subtracting the estimated baseline term (Eq. 11). The resulting reward signal more accurately reflects the true quality of generated prompts, enabling the policy to receive consistent feedback and perform more stable gradient updates. This empirical observation aligns with our theoretical analysis in Appendix A, confirming that the calibration effectively mitigates noise while preserving an unbiased reward estimate.

### D.6. Impact of Policy Agent Scale.

To investigate the impact of language model scale on prompt generation, we report additional results using the larger **Qwen2.5-7B-Instruct** (Yang et al., 2024) as the policy agent, comparing it against the default 3B version.

**Few-Shot Performance.** Table 18 presents the 16-shot classification results. CRL-BPT (7B) achieves an average accuracy of 64.1%, slightly outperforming the 3B variant (64.0%). While both versions rank first overall, they ex-

hibit complementary strengths: the 7B model excels on datasets requiring broad knowledge or reasoning (e.g., Caltech, SUN, and CLEVR), whereas the 3B model remains highly competitive on fine-grained tasks (e.g., Pets, Flowers, and Aircraft).

**Base-to-New Generalization.** Table 19 reports the base-to-new generalization results. CRL-BPT (7B) achieves 72.1% on base classes and 70.0% on new classes, yielding a harmonic mean of 70.6%. Compared to the 3B model, this represents a modest improvement of 0.5% in the harmonic mean, suggesting that increasing model scale contributes slightly to the generalizability of the generated prompts.

**Cross-Dataset Transfer and OOD Generalization.** Table 20 shows the cross-dataset transfer (CDT) and out-of-distribution (OOD) generalization performance. CRL-BPT (7B) attains 58.0% average accuracy on CDT (+0.8% over 3B) and 59.1% on OOD (+0.7% over 3B). This indicates that the reasoning capabilities of larger LLMs may facilitate better transfer to unseen domains and robust handling of distribution shifts.

**Summary.** Overall, scaling the policy agent from 3B to 7B parameters yields consistent but marginal improvements across all evaluation settings. This observation underscores the efficiency of CRL-BPT: our curriculum and stabilization mechanisms effectively guide the optimization, allowing even a compact 3B model to generate high-quality prompts that rival those from a larger model. While the strict budget of 2,000 queries might limit the exploration capacity of the 7B model, the results conclusively demonstrate that CRL-BPT achieves SOTA performance without relying on computationally expensive LLMs, making it a highly practical solution.

*Table 18.* Few-shot performance on 13 datasets. All the results are based on 16-shots per class. CRL-BPT (3B) uses Qwen2.5-3B-Instruct as the policy agent, while CRL-BPT (7B) uses Qwen2.5-7B-Instruct.

| Method | Caltech | Pets | Flowers | Food | Aircraft | SUN | DTD | SVHN | Cars | Resisc | CLEVR | UCF | ImageNet | Average |
|---|---|---|---|---|---|---|---|---|---|---|---|---|---|---|
| Zero-shot CLIP | 93.2 | 89.1 | 70.8 | 85.9 | 24.8 | 62.6 | 44.1 | 19.2 | 65.6 | 57.2 | 15.2 | 67.5 | 66.7 | 58.6 |
| BAR | 92.3 | 88.2 | 70.4 | 83.9 | 22.1 | 61.6 | 43.6 | 26.5 | 61.4 | 57.1 | 23.1 | 65.2 | 64.0 | 58.4 |
| BlackVIP | 92.4 | 87.7 | 68.4 | 82.8 | 22.2 | 61.2 | 41.6 | 22.7 | 61.8 | 55.0 | 25.2 | 64.8 | 63.5 | 57.7 |
| BPTVLM | 88.9 | 89.3 | 65.4 | 83.9 | 23.1 | 54.0 | 42.5 | 32.4 | 62.0 | 57.3 | 25.2 | 63.2 | 58.1 | 57.3 |
| ZIP | 92.1 | 91.2 | 67.5 | 86.0 | _25.3_ | 60.7 | 45.7 | 44.4 | _65.7_ | **63.1** | 21.2 | 68.2 | 64.1 | 61.2 |
| StablePrompt | 92.7 | 87.8 | 71.0 | 84.4 | 22.9 | 62.7 | 43.6 | 35.7 | 64.5 | 57.8 | 27.2 | 65.9 | 66.7 | 60.2 |
| CRL-BPT (3B) | _93.8_ | **92.1** | **74.1** | _86.4_ | **25.6** | _66.2_ | _47.0_ | _46.8_ | **66.2** | 62.4 | _32.3_ | **70.4** | _67.8_ | _64.0_ |
| CRL-BPT (7B) | **94.6** | _91.4_ | _73.5_ | **86.9** | 24.8 | **67.5** | **48.2** | **49.1** | 64.8 | 61.1 | **34.6** | _69.9_ | **68.2** | **64.1** |

*Table 19.* Base-to-new generalization performance.

| Method | Set | Caltech | Pets | Flowers | Food | Aircraft | SUN | DTD | SVHN | Cars | Resisc | CLEVR | UCF | ImageNet | Average |
|---|---|---|---|---|---|---|---|---|---|---|---|---|---|---|---|
| BAR | | 96.4 | 90.8 | 71.3 | 88.3 | 25.4 | 68.5 | 54.3 | 33.5 | 59.5 | 71.6 | 35.0 | 68.7 | 69.7 | 64.1 |
| BlackVIP | | 96.2 | 90.1 | 70.6 | 87.3 | 25.4 | 68.3 | 51.4 | 29.5 | 59.6 | 69.1 | 49.8 | 70.1 | 68.9 | 64.3 |
| BPTVLM | | 92.5 | 93.7 | 65.9 | 89.0 | 27.4 | 62.4 | 54.5 | 46.6 | 59.5 | 75.2 | 46.0 | 70.4 | 66.6 | 65.4 |
| ZIP | Base | 95.5 | 94.5 | 69.7 | _89.8_ | _30.3_ | 68.4 | **58.7** | 53.4 | **64.0** | 80.5 | 38.7 | _73.9_ | 71.3 | 68.4 |
| StablePrompt | | 95.7 | 92.1 | 68.3 | 88.4 | 28.8 | 69.8 | 55.9 | 49.2 | 62.2 | 72.4 | 55.4 | 69.2 | 72.5 | 67.7 |
| CRL-BPT (3B) | | **97.6** | 93.3 | **76.0** | 89.5 | **30.4** | _72.7_ | 57.3 | _62.3_ | _63.9_ | _79.7_ | _61.5_ | 73.2 | _73.5_ | _71.6_ |
| CRL-BPT (7B) | | _97.0_ | **96.7** | _75.4_ | **90.6** | 28.6 | **73.9** | _57.8_ | **63.1** | 63.8 | **80.6** | **62.4** | 73.0 | **74.1** | **72.1** |
| BAR | | **94.3** | 95.3 | 77.8 | 89.6 | 31.2 | 73.9 | 54.9 | 29.1 | 72.5 | 62.3 | 27.1 | 76.2 | 65.1 | 65.3 |
| BlackVIP | | 92.8 | 94.6 | 75.6 | 88.9 | 32.4 | 74.3 | **58.3** | 31.4 | 71.5 | 59.9 | 31.5 | 74.6 | 65.9 | 65.5 |
| BPTVLM | | 92.7 | 93.0 | 72.9 | 89.4 | 31.7 | 61.9 | 51.4 | 40.0 | 71.0 | 53.9 | 26.0 | 68.3 | 54.4 | 62.0 |
| ZIP | New | 93.3 | 96.2 | 72.1 | 89.7 | 30.2 | 70.6 | 50.4 | 42.2 | 72.8 | 63.1 | 26.3 | 69.6 | 65.1 | 64.7 |
| StablePrompt | | 93.3 | 95.4 | 76.8 | _90.1_ | 33.3 | 72.8 | 56.5 | 47.6 | _73.3_ | _65.9_ | 32.1 | 73.3 | 66.9 | 67.5 |
| CRL-BPT (3B) | | 93.8 | **97.8** | **78.9** | 90.5 | _35.1_ | 76.2 | 56.8 | _52.7_ | **73.8** | 65.9 | 35.8 | _77.7_ | 68.6 | _69.9_ |
| CRL-BPT (7B) | | _94.2_ | _97.3_ | 78.1 | 90.1 | **36.2** | **77.8** | _57.3_ | 53.4 | 73.0 | 66.8 | **36.7** | **78.6** | **68.9** | **70.0** |
| BAR | | 95.3 | 93.0 | 74.4 | 89.0 | 28.0 | 71.1 | 54.6 | 31.1 | 65.4 | 66.7 | 30.6 | 72.3 | 67.3 | 64.5 |
| BlackVIP | | 94.5 | 92.3 | 73.0 | 88.1 | 28.5 | 71.1 | 54.6 | 30.4 | 65.0 | 64.2 | 38.6 | 72.3 | 67.4 | 64.6 |
| BPTVLM | | 92.6 | 93.3 | 69.2 | 89.2 | 29.4 | 62.1 | 52.9 | 43.0 | 64.8 | 62.8 | 33.2 | 69.4 | 59.9 | 63.2 |
| ZIP | Harmonic | 94.4 | 95.4 | 70.9 | 89.7 | 30.3 | 69.5 | 54.2 | 47.1 | _68.2_ | 70.7 | 31.3 | 71.7 | 68.0 | 66.3 |
| StablePrompt | | 94.5 | 93.7 | 72.3 | 89.2 | 30.9 | 71.2 | 56.2 | 48.4 | 67.3 | 69.0 | 40.6 | 71.2 | 69.6 | 67.5 |
| CRL-BPT (3B) | | **95.7** | _95.5_ | 77.4 | _90.0_ | **32.6** | _74.4_ | _57.0_ | _57.1_ | 68.5 | _72.1_ | 45.3 | _75.3_ | 71.0 | _70.1_ |
| CRL-BPT (7B) | | _95.6_ | **97.3** | 76.7 | **90.3** | _32.0_ | **75.8** | **57.5** | **57.8** | 68.1 | **73.1** | **46.2** | **75.7** | **71.4** | **70.6** |

*Table 20.* Cross-dataset transfer and out-of-distribution generalization performance.

| | Source | CDT Target | | | | | | | | | | | | | OOD Target | | | | |
|---|---|---|---|---|---|---|---|---|---|---|---|---|---|---|---|---|---|---|---|
| Method | ImageNet | Caltech | Pets | Flowers | Food | Aircraft | SUN | DTD | SVHN | Cars | Resisc | CLEVR | UCF | Average | ImageNet-A | ImageNet-V2 | ImageNet-R | ImageNet-S | Average |
|---|---|---|---|---|---|---|---|---|---|---|---|---|---|---|---|---|---|---|---|
| BAR | 64.0 | 92.2 | _88.2_ | _70.4_ | 83.9 | 21.2 | 61.1 | 40.5 | 19.5 | 60.8 | 50.2 | _16.0_ | 64.8 | 55.8 | 40.2 | 57.5 | 71.8 | 43.8 | 53.3 |
| BlackVIP | 63.5 | 92.5 | 87.5 | 67.7 | 81.7 | **22.4** | 61.1 | 40.5 | 14.6 | 61.7 | 53.6 | **16.7** | 64.6 | 55.4 | 37.0 | 57.0 | 71.3 | 43.0 | 52.1 |
| BPTVLM | 58.1 | 80.2 | 74.3 | 48.9 | 79.9 | 19.4 | 47.1 | 32.9 | 15.8 | 56.6 | 43.6 | 14.9 | 55.3 | 47.4 | 37.3 | 49.1 | 66.0 | 35.0 | 46.9 |
| ZIP | 64.1 | 89.9 | 84.9 | 64.3 | 82.8 | 20.3 | 57.1 | 39.8 | 21.8 | 61.5 | 52.2 | 14.9 | 60.7 | 54.2 | 46.3 | 57.9 | 74.0 | 44.4 | 55.6 |
| StablePrompt | 66.4 | 92.5 | 82.1 | 68.8 | 83.1 | _22.2_ | 62.0 | **47.9** | 13.3 | 62.2 | 53.2 | 14.4 | 64.2 | 55.5 | 48.0 | 60.2 | 74.2 | 46.5 | 57.2 |
| CRL-BPT (3B) | _67.8_ | _93.2_ | 85.9 | **70.8** | _85.0_ | 21.4 | _64.5_ | _44.3_ | _24.1_ | **63.6** | _54.1_ | 12.8 | _67.1_ | _57.2_ | _49.6_ | _61.6_ | _75.3_ | _47.1_ | _58.4_ |
| CRL-BPT (7B) | **68.2** | **94.1** | **89.1** | 70.3 | **85.6** | 21.8 | **65.4** | 43.9 | **24.8** | _63.0_ | **56.5** | 14.3 | **67.5** | **58.0** | **50.1** | **62.1** | **76.2** | **47.9** | **59.1** |

