# OpenReview forum: "Curriculum Reinforcement Learning for Black-Box Prompt Tuning via Large Language Models"
_ICML.cc/2026/Conference — ICML 2026 regular_

### Official Review · Reviewer_VkWJ · 2026-02-16

**Soundness:** 3
**Presentation:** 3
**Significance:** 3
**Originality:** 2
**Overall Recommendation:** 4
**Confidence:** 4

**Summary:**

The paper introduces CRL-BPT, a curriculum reinforcement learning framework designed to optimize human-readable "hard" prompts for black-box VLMs like CLIP. Unlike existing methods that struggle with the "cold start" problem and high query costs, CRL-BPT utilizes a large language model as an agent and implements a dynamic curriculum that transitions from imitation to innovation. To ensure stable training under a limited API budget, the authors introduce tailored mechanisms like historical loss normalization and relative reward calibration. Extensive experiments across 13 datasets demonstrate that CRL-BPT achieves state-of-the-art accuracy and superior query efficiency, establishing a new benchmark for interpretable black-box prompt tuning.

**Compliance With Llm Reviewing Policy:**

Affirmed.

**Final Justification:**

The authors' rebuttal has fully resolved my main concerns, so I keep my overall assessment and incline to accept the paper.

**Key Questions For Authors:**

See weakness.

**Limitations:**

Yes

**Strengths And Weaknesses:**

Strength:
1. CRL-BPT achieves the highest average accuracy (64.0%) across 13 diverse datasets, ranging from general objects (ImageNet) to fine-grained classification (Flowers102) and specialized tasks (CLEVR). It consistently ranks first in 12 out of 13 datasets under a strict 2,000 API query budget, outperforming leading soft and hard prompt tuning baselines like ZIP and StablePrompt.

2. By implementing a transition from imitation (leveraging prior knowledge) to innovation (exploring novel patterns), the framework effectively bypasses the "cold start" problem that plagues typical reinforcement learning agents. Experimental trajectories show the model stabilizes in performance within just 800 queries, whereas versions without curriculum guidance fail to reach comparable accuracy even after the full 2,000 query budget.

Weakness

1. The reliance on ROUGE-L to calculate both imitation and innovation losses primarily captures lexical and structural overlaps. While superior to cosine similarity for structural tasks, this metric may fail to reward prompts that are semantically innovative but syntactically similar to the history buffer, potentially limiting the true diversity of discovered "hard" prompts.

2. While the decaying imitation loss is intended to prevent overfitting, the framework still relies on a manually curated Reference Set ($\mathcal{Z}$) for its early guidance. If the initial reference prompts are poorly chosen or too restrictive for a highly specialized domain, the model may converge to a suboptimal region of the search space before the "innovation" phase can effectively take over.

---

> ### Author Rebuttal · Authors · 2026-03-31
>
> Thank you for the constructive feedback. We address each point below.
>
> **W1 (ROUGE-L and true diversity):** We agree that **ROUGE-L mainly captures lexical and structural overlap**, and therefore is not a complete metric for semantic novelty. Our choice of ROUGE-L was motivated by an empirical observation in black-box VLM prompt engineering: prompts with very similar semantics can still lead to noticeably different downstream performance when their lexical or structural forms differ slightly. For example, in our additional analysis on EuroSAT, the prompts *“a photo of a hard to see {}.”* and *“a photo of the hard to see {}.”* have a cosine similarity of 99.46% under the CLIP text encoder, yet achieve 54.6% and 50.1% accuracy, respectively. This suggests that, in the BBPT setting for VLMs, **semantic similarity alone may not be sufficient** to capture the prompt patterns that matter for performance.
>
> For this reason, we use ROUGE-L as a lightweight structural signal in the imitation and innovation objectives, **rather than as the sole criterion for prompt quality**. This is also reflected in our overall objective,
>
> $$\mathcal{L}\_{\text{total}} = \widehat{\mathcal{L}}\_{\text{PPO}} + \alpha\_{\text{im}}(\tau) \widehat{\mathcal{L}}\_{\text{im}} + \alpha\_{\text{in}}(\tau) \widehat{\mathcal{L}}\_{\text{in}},$$
> where ROUGE-L only affects the auxiliary imitation/innovation terms, while prompt selection is ultimately driven by **PPO and task reward** from the black-box VLM. Therefore, **even if ROUGE-L does not explicitly reward every form of semantic novelty, the final optimization is not determined by ROUGE-L alone**. Empirically, this design is also supported by the ablation in **Appendix D.3**, where ROUGE-L outperforms cosine similarity computed via the CLIP encoder. At the same time, we agree that a hybrid metric combining **structural** and **semantic** similarity could be an interesting extension, and we will mention this limitation more clearly in the final version.
>
> **W2 (reliance on the reference set):** We agree that, in principle, a poorly chosen or overly restrictive reference set could bias early-stage exploration toward a suboptimal region. However, our additional analyses suggest that CRL-BPT is **not overly sensitive to the exact choice of the reference set**. Specifically, when we vary the reference pool from **Top-20** performing templates to **Top-10**, **Bottom-20**, or **Random-20**, the performance of  CRL-BPT degrades only gradually rather than collapsing, indicating that the method benefits from stronger references but does not critically depend on a narrowly optimized set.
>
> | Reference Set    | Flowers  |   SUN    |   UCF    |   Avg.   |
> | :--------------- | :------: | :------: | :------: | :------: |
> | Top-1            |   72.8   |   65.4   |   69.3   |   69.1   |
> | Top-10           |   73.4   |   65.8   |   69.6   |   69.6   |
> | Top-20 (default) | **74.1** | **66.2** | **70.4** | **70.2** |
> | Top-40           |   73.4   |   65.1   |   69.9   |   69.5   |
> | Bottom-20        |   72.5   |   65.7   |   68.1   |   68.7   |
> | Random-20        |   73.0   |   65.5   |   67.8   |   68.8   |
>
> Moreover, the imitation signal in CRL-BPT is **explicitly decayed over training**, while the innovation term is progressively strengthened. Therefore, the framework is designed precisely to reduce long-term dependence on the initial references, allowing the policy to move beyond the early prior instead of remaining trapped by it.

---

> > ### Author Rebuttal · Reviewer_VkWJ · 2026-03-31
> >
> > Thank you for additional results and clarification. With the new results, I believe my assessment is correct and I will raise my confidence.

---

> > > ### Author Response · Authors · 2026-04-03
> > >
> > > Thank you for the positive feedback. We are glad that the additional results and clarifications addressed your concerns, and we appreciate your increased confidence in our work.

---

### Official Review · Reviewer_YCuX · 2026-03-13

**Soundness:** 3
**Presentation:** 3
**Significance:** 2
**Originality:** 2
**Overall Recommendation:** 4
**Confidence:** 3

**Summary:**

This paper proposes CRL-BPT, a curriculum RL framework that teaches an LLM as a policy agent to generate textual prompts for black-box VLMs. The method adds two auxiliary losses to RL training, including an imitation loss and an innovation loss. Two stabilization mechanisms were also proposed to address training instability. The paper reports state-of-the-art results across 13 image classification benchmarks under a relatively low API call budget, demonstrating simultaneous gains in accuracy, query efficiency, and prompt interpretability.

**Compliance With Llm Reviewing Policy:**

Affirmed.

**Final Justification:**

All my questions are resolved

**Key Questions For Authors:**

Questions:
1. The role of reference set in RL training is under-explored. How is the quality measured? What if only half of the reference prompts are used (will this degrade performance)? What if the high performing prompts obtained after one round optimization is included (will this improve performance)?
2. Were any of the experiments run with multiple times to compute the standard deviation such that the stability of the approach can be estimated?
3. The definition of curriculum learning appears different from usual understanding where the training samples are scheduled instead of randomly shuffled. To claim contribution in curriculum learning, has the paper explored alternative curriculum-based learning approaches such as encouraging the model to learn from easy to hard, or learning mostly from samples with intermediate difficulty?

**Limitations:**

Limitations are not explicitly discussed.

**Strengths And Weaknesses:**

Strengths:
1. The paper is well written and clearly motivated
2. The design of the loss terms are reasonable and principled.
3. Comprehensive evaluation shows that the model is a strong compared to other baseline approaches in terms of effectiveness and sample efficiency.
4. Ablation studies show the effectiveness of each individual components.

Weaknesses:
1. The “cold start” problem for RL appears to be not well-defined. Is it a common notion? It was not mentioned in Deng et al., 2022, but the paper said “often referred to” on line 111.
2. The usage of ROUGE score presumes that it is well aligned with human judgements, but using LLM-as-a-judge could be more accurate.
3. The reference set (of prompts) plays an important role here. However, there is a lack of study on how does the quantity/quality of these prompts influence the training.

---

> ### Author Rebuttal · Authors · 2026-03-31
>
> Thank you for the constructive feedback. We address each point below.
>
> **W1 (cold start):**  We agree that the phrase “often referred to as the cold start problem” is imprecise. Our intention was not to claim that Deng et al. (2022) explicitly defines a standard “cold start” notion, but to use the term **informally to describe the early stage** of RL-based hard prompt optimization in a large discrete search space, where the policy lacks useful priors and therefore performs largely blind exploration, wasting many queries on low-quality prompts. We will revise the wording accordingly to avoid overstating the terminology.
>
> **W2 (ROUGE vs. LLM-as-a-judge):** We agree that LLM-as-a-judge can better approximate **human semantic judgments**, but this is not the role of ROUGE-L in our method. Here, the similarity metric serves as a **low-cost, deterministic training signal** for the imitation/innovation losses, and **must be computed at every RL step** under a strict query budget. ROUGE-L is suitable because it captures lexical/structural patterns in reference prompts, which is exactly what the imitation objective encourages. By contrast, using an additional LLM judge would **increase query cost and may introduce extra stochasticity into training**. In **Appendix D.3**, our ablation also shows that ROUGE-L outperforms cosine similarity, which supports its effectiveness in our current setting. We will consider LLM-as-a-judge as a promising future direction.
>
> **W3&Q1 (role of the reference set):** As stated in **Appendix C.3**, reference quality is measured by the prompt’s performance on 200 training batches, and the default reference set is constructed by selecting the top-20 performing prompts from CLIP’s 80 manual templates. We further conducted a sensitivity study using **Top-k, Bottom-k, and Random-k** reference sets. As show below, performance degrades gracefully when fewer or lower-quality references are used: using Top-10 is slightly worse than Top-20, while Bottom-20 and Random-20 still remain competitive. This suggests that CRL-BPT benefits from stronger references, but is **not highly sensitive to the exact choice of the reference set**.
>
> | Reference Set    | Flowers  |   SUN    |   UCF    |   Avg.   |
> | :--------------- | :------: | :------: | :------: | :------: |
> | Top-1            |   72.8   |   65.4   |   69.3   |   69.1   |
> | Top-10           |   73.4   |   65.8   |   69.6   |   69.6   |
> | Top-20 (default) | **74.1** | **66.2** | **70.4** | **70.2** |
> | Top-40           |   73.4   |   65.1   |   69.9   |   69.5   |
> | Bottom-20        |   72.5   |   65.7   |   68.1   |   68.7   |
> | Random-20        |   73.0   |   65.5   |   67.8   |   68.8   |
>
> We also tested augmenting the reference set with high-performing prompts obtained after one optimization round:
>
> | Reference Set       | Flowers  |   SUN    |   UCF    |   Avg.   |
> | :------------------ | :------: | :------: | :------: | :------: |
> | CRL-BPT (default)   | **74.1** | **66.2** | **70.4** | **70.2** |
> | CRL-BPT (augmented) |   72.8   |   66.0   |   68.8   |   69.2   |
>
> Interestingly, **augmentation does not improve performance**; instead, the average accuracy drops from **70.2%** to **69.2%**. This suggests that the reference set mainly acts as a **warm-start prior** for imitation, rather than a pool that should be iteratively refined with self-generated prompts. A plausible explanation is that overly strong or highly similar references can bias the policy toward a narrow region of the prompt space during imitation, thereby reducing the benefit of subsequent innovation.
>
> **Q2 (stability across runs):**  Yes. All reported results were averaged over **three random seeds**. Following prior BBPT works such as ZIP and BPTVLM, we reported the averaged results in the main tables for consistency. Following your suggestion, we additionally report the corresponding **mean ± standard deviation** on representative datasets below, which confirms that CRL-BPT is stable across runs.
>
> | Method  |  Flowers   |    SUN     |    UCF     |
> | :------ | :--------: | :--------: | :--------: |
> | CRL-BPT | 74.1 ± [0.6] | 66.2 ± [0.4] | 70.4 ± [0.7] |
>
> **Q3 (curriculum learning formulation):** We agree that curriculum learning is usually understood as **sample-level scheduling** (e.g., easy-to-hard samples). However, our method uses curriculum **at the objective level**, by gradually shifting training from reference-guided imitation to history-aware innovation through a dynamic weighting schedule. As discussed in **Related Work**, our contribution is therefore an **objective-based curriculum**, not a conventional sample-ordering curriculum. This is supported by our ablations (**Section 5.3**): removing imitation/innovation hurts performance, and linear/constant schedules underperform the proposed exponential schedule. The sample-level alternatives are valuable and largely orthogonal to our design, and exploring them in BBPT would be an interesting direction for future work.

---

> > ### Author Rebuttal · Reviewer_YCuX · 2026-04-03
> >
> > Thank the authors for adequately addressing my concerns. I will maintain the score for acceptance.

---

> > > ### Author Response · Authors · 2026-04-03
> > >
> > > Thank you for your careful consideration of our rebuttal. We are very pleased that our response has adequately addressed your concerns. We sincerely appreciate your positive feedback and your support for acceptance.

---

### Official Review · Reviewer_pnJL · 2026-03-17

**Soundness:** 2
**Presentation:** 2
**Significance:** 2
**Originality:** 2
**Overall Recommendation:** 4
**Confidence:** 3

**Summary:**

Black-box prompt tuning (BBPT) aims to optimize input prompts for large models where internal parameters and gradients are inaccessible.
Existing BBPT methods are generally categorized into continuous soft prompt tuning and discrete hard prompt tuning.
Soft prompt tuning optimizes continuous embedding vectors via derivative-free methods, which can be effective but generally suffers from slow convergence in high-dimensional parameter spaces and lacks interpretability, as the learned prompts are opaque vectors.
On the other hand, hard prompt tuning optimizes discrete, human-readable prompt tokens.
Several approaches utilize an LLM as a policy agent and formulate prompt generation as an RL problem.
However, RL-based approaches suffer from the cold start exploration problem, where the agent wastes thousands of API calls exploring low-quality prompts before discovering useful patterns.

To address these challenges, this paper proposes Curriculum Reinforcement Learning for Black-box Prompt Tuning via LLMs (CRL-BPT) by incorporating a curriculum learning strategy and training stabilization mechanisms.
For the curriculum learning, the authors introduce an imitation loss that encourages the agent to imitate high-quality reference prompts and an innovation loss that promotes exploration by penalizing similarity to previously generated prompts.
In curriculum learning strategy, a dynamic curriculum schedule is designed to facilitate a smooth transition from imitation to innovation during training.
These losses are then integrated with the standard PPO optimization.
To stabilize training, the authors propose two additional approaches: (1) historical loss normalization, which rescales each loss term using an Exponential Moving Average (EMA) to balance their magnitudes, and (2) relative reward calibration, which reduces reward variance by calibrating reward based on a baseline prompt evaluated on the same batch.

In the experimental section, the authors compare CRL-BPT against three categories of baselines: (1) hand-crafted prompts, (2) soft prompt tuning methods, and (3) hard prompt tuning methods.
CRL-BPT outperforms prior approaches on 12 out of 13 datasets under a strict query budget ( = 2,000 API calls), demonstrating improvements in query efficiency, RL training stability, and generalization while maintaining interpretability.

**Compliance With Llm Reviewing Policy:**

Affirmed.

**Final Justification:**

The authors provide detailed response and the additional empirical evaluation.

**Key Questions For Authors:**

- The method relies on a reference prompt set constructed from top-20 performing hand-crafted prompts. How sensitive is the performance to the quality of these reference prompts?
- It is unclear whether the method truly discovers novel prompts or simply selects and paraphrases prompts from the reference set. As a simple upper-bound baseline, could the authors report the performance of selecting the best-performing prompt from the reference set (e.g., max over reference prompts at each evaluation step)?
- It would be important to evaluate additional RL-based hard prompt tuning baselines using the same policy model (i.e., Qwen2.5-(3B/7B)-Instruct) to ensure a fair comparison, particularly in terms of query efficiency (with 2,000 API calls) and final performance (with larger budgets).

**Strengths And Weaknesses:**

### Strengths
- This paper addresses two key challenges in BBPT (query efficiency and prompt interpretability) in a unified framework, CRL-BPT.
- The proposed curriculum learning strategy is intuitive and practically motivated and it effectively guides the optimization from imitation to innovation.
- CRL-BPT demonstrates strong empirical performance across a wide range of benchmarks, outperforming prior prompt tuning approaches under a strict query budget.

### Weaknesses
- The overall novelty is somewhat limited. CRL-BPT mainly combines existing components (e.g., imitation loss, innovation loss, curriculum scheduling, and variance reduction) rather than introducing fundamentally new ideas.
- Many design choices are heuristic and lack theoretical justification (including the dynamic curriculum schedule, similarity metric in imitation loss, and novelty scores in innovation loss).
- While the paper criticizes existing RL-based hard prompt tuning methods, the experimental evaluation includes only one RL-based approach (**StablePrompt**), raising concerns about the fairness and completeness of the comparison.

Overall, while this paper introduces a practically well-engineered approach, the lack of novelty and reliance on heuristics limit its contribution.

---

> ### Author Rebuttal · Authors · 2026-03-31
>
> Thank you for the constructive feedback. We address each point below.
>
> **W1 (novelty):** We would like to clarify that the novelty of CRL-BPT lies in a **unified objective-level curriculum RL framework** for BBPT of VLMs that jointly improves **prompt interpretability** and **query efficiency** under a strict API budget. Specifically, CRL-BPT guides the LLM policy from **reference-guided imitation** to **history-aware innovation**, unlike prior curriculum methods that mainly use sample-level scheduling. To our knowledge, **prior BBPT methods do not explicitly combine these two goals in such a framework**.
>
> **W2 (heuristic design/theory):** Our theoretical claim focuses on **relative reward calibration (Appendix A)**: the calibrated reward is unbiased, and the optimal coefficient is analytically derived to minimize variance. The curriculum schedule, similarity metric, and novelty score are **task-specific design choices**, not claimed to be theoretically optimal. We validate them empirically, as shown by the ablations in **Section 5.3 and Appendix D**.
>
> **W3 (fairness of RL comparisons):** Since most RL-based methods were originally developed for NLP, we initially included the strongest adapted baseline, StablePrompt. Following your suggestion, we further reproduced **RLPrompt** and **TEMPERA** under the same BBPT setting for VLMs.  As shown below, CRL-BPT consistently outperforms all RL baselines, suggesting that the gain comes from our curriculum design and stabilization mechanisms, rather than RL optimization alone.
>
> | Method       | Caltech  |   Pets   | Flowers  |   Food   | Aircraft |   SUN    |   DTD    |   SVHN   |   Cars   |  Resisc  |  CLEVR   | UCF      | ImageNet |   Avg.   |
> | :----------- | :------: | :------: | :------: | :------: | :------: | :------: | :------: | :------: | :------: | :------: | :------: | -------- | :------: | :------: |
> | RLPrompt     |   92.8   |   86.9   |   69.2   |   83.2   |   23.4   |   61.9   |   43.4   |   29.6   |   63.9   |   56.7   |   19.4   | 65.6     |   66.1   |   58.6   |
> | TEMPERA      |   93.5   |   88.1   |   70.2   |   84.2   |   21.8   |   61.9   |   44.5   |   31.2   |   63.7   |   58.1   |   16.8   | 66.7     |   66.8   |   59.0   |
> | StablePrompt |   92.7   |   87.8   |   71.0   |   84.4   |   22.9   |   62.7   |   43.6   |   35.7   |   64.5   |   57.8   |   27.2   | 65.9     |   66.7   |   60.2   |
> | CRL-BPT      | **94.6** | **92.1** | **74.1** | **86.4** | **25.6** | **66.2** | **47.0** | **46.8** | **66.2** | **62.4** | **32.3** | **70.4** | **67.8** | **64.0** |
>
> **Q1 (sensitivity to reference prompts):** We analyzed **Top-k, Bottom-k, and Random-k reference sets**. As shown below, performance degrades gracefully with fewer or lower-quality reference prompts. Importantly, **even with Bottom-20 or Random-20 references**, CRL-BPT still outperforms StablePrompt on the same three datasets by **over 2 points on average**. Thus, CRL-BPT benefits from better references, but is **not highly sensitive to the exact choice of the reference set**.
>
> | Reference Set    | Flowers  |   SUN    |   UCF    |   Avg.   |
> | :--------------- | :------: | :------: | :------: | :------: |
> | Top-1            |   72.8   |   65.4   |   69.3   |   69.1   |
> | Top-10           |   73.4   |   65.8   |   69.6   |   69.6   |
> | Top-20 (default) | **74.1** | **66.2** | **70.4** | **70.2** |
> | Top-40           |   73.4   |   65.1   |   69.9   |   69.5   |
> | Bottom-20        |   72.5   |   65.7   |   68.1   |   68.7   |
> | Random-20        |   73.0   |   65.5   |   67.8   |   68.8   |
>
> **Q2 (novel discovery vs. paraphrasing):** CRL-BPT goes beyond merely copying or paraphrasing the references. We report the zero-shot performance of the **best-performing reference prompt** (Avg. 62.2%). For example, on Flowers, the best reference prompt is *“a close-up photo of a {}.”* (71.2%), whereas CRL-BPT discovers *“Lovely sunny day, capture a beautiful {} blooming beautifully in the garden”* (74.1%). As further shown in **Appendix D.1**, the learned prompts often introduce **dataset-specific semantics** that are absent from the reference templates.
>
> **Q3 (same policy model):** Following your suggestion, we compared all RL-based methods **using the same policy model**. The strict 2,000 API-call results are reported in the W3 table, while the fully converged results are listed below. CRL-BPT achieves both the **highest final accuracy** and the **fewest API calls**, indicating that the gain is **not due to the policy model alone**.
>
> | Method       | Flowers  |   SUN    |   UCF    |   Avg.   | API Calls  |
> | :----------- | :------: | :------: | :------: | :------: | :--------: |
> | RLPrompt     |   72.4   |   64.2   |   68.2   |   68.3   |  ~20,000   |
> | TEMPERA      |   71.9   |   63.5   |   69.1   |   68.2   |  ~12,900   |
> | StablePrompt |   73.2   |   65.9   |   67.2   |   68.8   |   ~8,300   |
> | CRL-BPT      | **74.9** | **67.4** | **71.3** | **71.2** | **~5,000** |

---

> > ### Author Rebuttal · Reviewer_pnJL · 2026-04-08
> >
> > Thank you for your detailed response and the additional empirical evaluation.
> > I will raise my score to weak accept.

---

> > > ### Author Response · Authors · 2026-04-08
> > >
> > > Thank you very much for your time and for carefully considering our rebuttal.
> > > We truly appreciate your constructive feedback and are glad that our clarifications and additional experiments helped address your concerns.
> > > Thank you again for your valuable comments and support.

---

### Decision · Program_Chairs · 2026-04-30

**Decision:**

Accept (regular)

**Comment:**

This paper introduces CRL-BPT, a curriculum reinforcement learning framework for black-box prompt tuning that optimizes human-readable prompts for VLMs. During the discussion phase, the authors successfully addressed concerns regarding the "cold start" problem in reinforcement learning and provided a comprehensive sensitivity analysis demonstrating that the method maintains consistent gains regardless of the reference set quality. While all reviewers ultimately converged on a recommendation for acceptance, none emerged as a strong champion for the work, primarily due to the limited technical novelty as the framework is largely seen as a well-engineered combination of existing components and a reliance on heuristics like ROUGE-L for structural similarity. Nevertheless, the AC belives that the paper is technically solid, features extensive experimental validation, and likely offers practical value for the ICML audience interested in efficient and interpretable prompt optimization.